# Contributions of Transported Prudhoe Bay Oil Field Emissions to the Aerosol Population in Utqiaġvik, Alaska

Matthew J. Gunsch[1], Rachel M. Kirpes[1], Katheryn R. Kolesar[1], Tate E. Barrett[2], Swarup China[3], Rebecca J. Sheesley[2,4], Alexander Laskin[3], Alfred Wiedensohler[5], Thomas Tuch[5] Kerri A. Pratt[1,6]

[1]Department of Chemistry, University of Michigan, Ann Arbor, MI, USA
[2]The Institute of Ecological, Earth, and Environmental Sciences, Baylor University, Waco, TX, USA
[3]Environmental Molecular Sciences Laboratory, Pacific Northwest National Laboratory, Richland, WA, USA
[4]Department of Environmental Science, Baylor University, Waco, TX, USA
[5]Leibniz Institute for Tropospheric Research, Leipzig, Germany
[6]Department of Earth and Environmental Sciences, University of Michigan, Ann Arbor, MI, USA

*Correspondence to:* Kerri A. Pratt (prattka@umich.edu)

**Abstract**. Loss of sea ice is opening the Arctic to increasing development involving oil and gas extraction and shipping. Given the significant impacts of absorbing aerosol and secondary aerosol precursors emitted within the rapidly warming Arctic region, there is a need to characterize local anthropogenic aerosol sources and compare to natural conditions. From August-September 2015 in Utqiaġvik, AK, the chemical composition of individual atmospheric particles was measured by computer-controlled scanning electron microscopy with energy dispersive X-ray spectroscopy (0.13 - 4 µm projected area diameter) and real-time single particle mass spectrometry (0.2 – 1.5 µm aerodynamic diameter). During Arctic Ocean influenced periods (70% of the study), our results show that fresh sea spray aerosol contributed ~20%, by number, of particles between 0.13 – 0.4 µm, 40 – 70% between 0.4 – 1 µm, and 80 – 100% of 1 – 4 µm particles. In contrast, for periods influenced by emissions from Prudhoe Bay (10% of the study), the third largest oil field in North America, there was a strong influence from submicron (0.13 – 1 µm) combustion derived particles (20 - 50% OC, by number, 5 – 10% soot by number). While sea spray aerosol still comprised a large fraction of particles (90% by number from 1 – 4 µm) detected under Prudhoe Bay influence, these particles were internally mixed with sulfate and nitrate indicative of aging processes during transport. In addition, the overall mode of the particle size number distribution shifted

from 76 nm during Arctic Ocean influence to 27 nm during Prudhoe Bay influence with particle concentrations increasing from 130 cm$^{-3}$ to 920 cm$^{-3}$ due to transported particle emissions from the oil fields. The increased contributions of carbonaceous combustion products and partially aged sea spray aerosol should be considered in future Arctic atmospheric composition and climate simulations.

# 1 Introduction

The Arctic is experiencing dramatic climate change with sea ice extent declining rapidly and complete summertime sea ice loss expected by 2050 (Wang and Overland, 2015; Overland and Wang, 2013). With 30% of the world's undiscovered natural gas and 13% of undiscovered oil thought to be located in the Arctic (Gautier et al., 2009), increasing open water makes previously inaccessible areas of the Arctic available for oil and gas development and shipping (Harsem et al., 2015; Allison and Bassett, 2015). These oil and gas extraction activities add pollutants, including particulate matter (PM), volatile organic compounds (VOCs), $SO_2$, and $NO_x$, to the Arctic atmosphere (Peters et al., 2011), thereby influencing climate. The Arctic aerosol population is characterized by a maximum mass loading in the winter, due to transported pollutants from the mid-latitudes, and a minimum in the summer, when local sources, including sea spray aerosol, dominate (Quinn et al., 2002; Quinn et al., 2007). However, there is limited knowledge of aerosols produced within the Arctic, particularly in the context of changing emissions from both natural and anthropogenic sources (Arnold et al., 2016).

Modeling by Peters et al. (2011) estimates that Arctic oil and gas extraction during 2004 contributed 47 kilotons (kt) of PM emissions, with 15 kt corresponding to black carbon (BC) and 16 kt attributed to organic carbon (OC). The majority of emissions originated in western Russia (~41 kt in 2004); activities within the Alaskan Arctic, primarily the Prudhoe Bay oil fields, contributed 6 kt during 2004 (Peters et al., 2011). Prudhoe Bay is the third largest oil field and tenth largest gas field in the US by estimated production as of 2013 (EIA, 2015). The majority of PM emitted by US Arctic oil and gas extraction sources (turbine gas combustion, diesel emissions from generators and vehicles, and flaring (Stohl et al., 2013)) in 2004 corresponded to BC (1.9 kt) and OC (2.0 kt) (Peters et al., 2011). With new drilling operations opening due to reduced sea ice coverage, Peters et al. (2011) estimate US contributions increasing up to 10 kt of primary PM (including 3.3 kt BC and 3.5 kt OC) by 2030 and 17 kt of PM (including 5.3 kt BC and 5.7 kt OC) by 2050. In addition to directly emitted PM, drilling operations can emit aerosol precursors ($NO_x$, $SO_2$, and VOCs), and alter oxidant levels, which can lead to the formation of secondary aerosol, as well as contribute to new particle formation (Peters et al., 2011; Volkamer et al., 2006; Roiger et al., 2015; Kolesar et al., 2017; Jaffe et al., 1995).

BC is estimated to have a warming effect on the Arctic atmosphere (e.g. Flanner, 2013; Sand et al., 2013a; Sharma et al., 2013; Bond et al., 2013; Flanner et al., 2007). Modeling predicts local Arctic BC emissions to cause as much as a factor of five greater increase in Arctic warming compared to BC transported from the lower latitudes (Sand et al., 2013b). Koch et al. (2009) suggest that Arctic BC concentrations are under-predicted by a variety of models by an average factor of 2.5, which may be improved by more accurately incorporating local BC sources (Flanner, 2013). However, few studies have measured PM emitted from oil and gas extraction in the Arctic. Measurements of BC from 1977 – 1997 in Utqiaġvik showed contributions from Russian oil fields year round (Polissar et al., 1999; Polissar et al., 2001), similar to the recent results of Barrett et al. (2015) at Utqiaġvik during December 2012 – March 2013 when transported particles from Russian oil fields were observed. Barrett et al. (2015) also measured regional Arctic BC from both fossil fuel combustion and biomass burning. The Arctic Climate Change, Economy, and Society (ACCESS) aircraft field campaign recently investigated emissions from oil and gas extraction in the Norwegian Arctic and measured increased BC, among other pollutants, compared to the Arctic background while sampling within plumes from oil and gas extraction facilities (Roiger et al., 2015). Also, Brock et al. (2011) conducted aircraft measurements of Prudhoe Bay emissions and detected increased PM, including OC and BC. Stohl et al. (2013) modeled BC contributions from flaring due to Arctic oil and gas extraction and determined that it contributed 42% of the annual surface soot concentrations in the Arctic. With only these limited measurements available, further characterization of combustion emissions from oil and gas extraction activities are needed to further improve simulations.

During transport, primary aerosol can undergo chemical aging and accumulate secondary species, such as sulfate, nitrate, ammonium, oxidized organic carbon, and water, impacting both chemical composition and particle properties, such as light absorption and scattering, hygroscopicity, toxicity, and chemical reactivity (Moffet and Prather, 2009; Pöschl, 2005). Combustion particles in particular are co-emitted with VOCs and can rapidly undergo aging and accumulate these secondary organic species (Petzold et al., 2005). The distribution of these secondary species across the aerosol population determines aerosol climate impacts (Prather et al., 2008). It is currently not clear whether light absorption by a BC particle is enhanced by sulfate or organic coatings (e.g., Chung and Seinfeld, 2005; Jacobson, 2001; Moffet and Prather, 2009; Knox et al., 2009; Liu et al., 2015; Cappa et al., 2012; Healy et al., 2015). In

contrast, pure sulfate particles (i.e., without BC (soot) inclusions) primarily scatter light (Haywood and Boucher, 2000). In addition, soot particles internally mixed with nitrate and sulfate have been shown to have increased CCN-activity (Bond et al., 2013). Ambient aerosol populations typically vary between internal mixtures, with multiple chemical species contained within a single particle, and external mixtures, with multiple chemical species present as separate particles (Prather et al., 2008). Therefore, it is important to determine the influence of Prudhoe Bay emissions on downwind aerosol chemical composition and mixing state of the individual particles in order to better understand and predict the effects of oil and gas extraction activities on the Arctic aerosol population and climate.

To investigate particle chemical composition and sources in the coastal Alaskan Arctic, sampling was conducted at Utqiaġvik, Alaska, a location influenced by Prudhoe Bay (Jaffe et al., 1995) and the Arctic Ocean (Quinn et al., 2002) in August – September 2015. On-line aerosol time-of-flight mass spectrometry (ATOFMS) and off-line computer controlled scanning electron microscopy with energy dispersive X-ray spectroscopy (CCSEM-EDX) analyses provided size-resolved individual particle chemical composition. The impacts of transported Prudhoe Bay oil field emissions on aerosol size distributions, primary combustion particle contributions, and secondary aerosol formation are compared to the background Arctic aerosol combustion.

## 2 Methods

Atmospheric aerosol sampling was conducted over a period of August 21 – September 30, 2015 at a field site (71°16'30"N, 156°38'26"W), on the Barrow Environmental Observatory, located 4 km southeast of the town of Utqiaġvik, AK. Aerosol sampling occurred 4.5 m above ground level through 1.4 cm ID copper tubing at a flow rate of 17 Lpm through a $PM_{10}$ (PM less than 10 μm) teflon-coated aluminum cyclone (URG-2000-30ENB, URG Corp., Chapel Hill, NC). A stainless steel cylindrical manifold (ID 8.9 cm) split the flow to dedicated insulated sampling lines for each instrument. Meteorological data, including wind speed, wind direction, relative humidity, and temperature, were obtained from the National Oceanic and Atmospheric Administration (NOAA) Earth System Research Laboratory (ESRL) Global Monitoring Division (GMD) long-term monitoring station (NOAA Barrow Observatory, 71° 19'

40" N, 156° 38' 20" W), located 5.5 km across flat tundra to the northeast of the aerosol sampling site. On-line measurements of aerosol absorption at seven wavelengths, including 880 nm, were completed using an aethalometer to obtain BC mass concentrations. BC concentrations were calculated, at 5-10 minute time resolution, using a portable aethalometer (Model AE42, Magee Scientific, Berkeley, CA). The aethalometer was outfitted with a seven wavelength source and PM$_{2.5}$ inlet. The sample is collected on a quartz fiber filter tape and the optical analysis (wavelengths ranging from 370 to 950 nm) is performed continuously.

## 2.1 Air Mass Classification

Backward air mass trajectories were calculated using the NOAA Hybrid Single Particle Lagrangian Integrated Trajectory (HYSPLIT) Model (Stein et al., 2015). A final height of 50 m AGL was used for arrival at the field site, and a new trajectory was calculated every 8 h and modeled the preceding 48 h. Trajectories originated from three major directions: north/northeast, southeast, and to the west (Figure 1). Based on these trajectories, air masses were classified into three areas of influence. Air masses that originated in the Beaufort Sea to the north/northeast of the site were classified as Arctic Ocean influenced periods. Air masses from the southeast that crossed over the Prudhoe Bay oil fields, were classified as Prudhoe Bay influenced air masses, based on a previous study of air mass transport from Prudhoe Bay to Utqiaġvik (Kolesar et al., 2017). Air masses that originated from the west were primarily influenced by the town of Utqiaġvik and classified as local influence.

## 2.2 Particle Number Distributions

A scanning mobility particle sizer (SMPS, model 3082, TSI Inc., Shoreview, MN) was located at the field site from August 21 – September 20, 2015 for online measurements of size-resolved particle number concentrations from 13 - 746 nm (mobility diameter). Additionally, long-term measurements of particle number size distributions from the NOAA Barrow Observatory were collected with a TROPOS-type mobility particle size spectrometer (Wiedensohler et al., 2012) to determine daily average particle number concentrations and size distributions in August and September for the available years of 2008, 2009, 2013, and 2014. Time periods when the wind direction was between 170 and 330° were excluded from

daily averages because of the influence from the town of Utqiaġvik. In addition, short (< 1 h) bursts of ultrafine particles during clean time periods were excluded due to the likely short-term influence from local vehicle emissions. For the long-term data, the daily averages were classified according to air mass source region using 48 h backward air mass trajectories and then averaged over a month-long period.

## 2.3 Computer-Controlled Scanning Electron Microscopy with Energy Dispersive X-Ray Spectroscopy

From August 21-September 30, 2015, 0.07 – 5.0 μm particles were collected during 8 h sampling periods (00:00 – 08:00, 08:00 – 16:00, 16:00 – 00:00 AKDT) on aluminum foil substrates (MSP Corp., Shoreview, MN) and transmission electron microscopy (TEM) grids (carbon Type-B Formvar film copper grids, Ted Pella Inc., Redding, CA) using a three-stage impactor (MPS-3, California Measurements, Sierra Madre, CA) with aerodynamic diameter size cuts of 0.07 – 0.4 μm, 0.4 –2.8 μm, and 2.8 – 5 μm, respectively. Individual particles were analyzed using computer-controlled scanning electron microscopy with energy dispersive X-ray spectroscopy (CCSEM-EDX). A FEI Quanta environmental SEM with a field emission gun (FEG) operating at 20 keV with a high-angle annular dark field (HAADF) detector collected SEM images and morphological data (including diameter, perimeter, and projected area) of individual particles 0.13 – 4.0 µm projected area diameter (Laskin et al., 2006; Laskin et al., 2012). The instrument is equipped with an EDX spectrometer (EDAX Inc., Mahwah, NJ) to measure X-ray spectra of elements with atomic numbers higher than Be, providing the relative atomic abundance of elements C, N, O, Na, Mg, Al, Si, P, S, Cl, K, Ca, Ti, V, Fe, and Zn. Additional CCSEM-EDX analysis was conducted using the same method with an FEI Helios 650 Nanolab SEM/FIB (focused ion beam) with an FEG operating at 20 keV using HAADF and through-the-lens detectors with an EDX spectrometer.

K-means cluster analysis was conducted over EDX spectra from 13,972 individual particles analyzed by the Quanta instrument and 5,121 particles analyzed by the Helios instrument, resulting in 50 clusters from each data set. These clusters were then regrouped into seven main particle classes based on elemental composition, described in Section 3.2 and the supplemental information. For periods corresponding to Arctic Ocean air mass influence, SEM images and EDX spectra were obtained for 2,869

particles from four samples which coincided with ATOFMS sampling: September 8, 2015 (00:00 – 08:00 and 08:00 – 16:00 AKDT), September 9, 2015 00:00 – 08:00, and September 15, 2015 00:00 – 08:00. For periods of Prudhoe Bay air mass influence, 1,997 particles from two samples which coincide with ATOFMS sampling (September 23, 2015 00:00 – 08:00 and 08:00 – 16:00) were analyzed. Error resulting from number fraction for different particle types were calculated using binomial statistics, and the minimum number of particles for a representative sample are between ~300 - 1,000 (Willis et al., 2002).

## 2.4 Aerosol Time-of-Flight Mass Spectrometer (ATOFMS)

An aerosol time-of-flight mass spectrometer (ATOFMS) measured the size and chemical composition of individual aerosol particles (0.2-1.5 μm) in real-time from September 8 – 30, 2015. The ATOFMS used in the current study is based on the design of Pratt et al. (2009) with modifications as described below. Briefly, particles are focused using an aerodynamic lens system, and particle velocity is measured by the transit time between two continuous wave lasers, 405 nm and 488 nm (OBIX LX, Coherent, Inc., Santa Clara, CA), spaced 6 cm apart. Vacuum aerodynamic particle diameter ($d_{va}$) is calculated based on particle velocity from polystyrene latex sphere standards of known diameter (90 nm – 1.5 μm) and density (1 g cm$^{-3}$). Particles enter a dual-polarity reflectron time-of-flight mass spectrometer (Tofwerk AG, Thun, Switzerland) and are desorbed and ionized by a Q-switched 100 Hz 266nm Nd:YAG laser (Centurion, Quantel USA, Bozeman, MT) operated at 0.8 – 1.0 mJ, resulting in positive and negative ion mass spectra of laser-ablated individual particles. Mass spectral peak lists were created in custom software developed in LabVIEW and MATLAB. Prior to ATOFMS sampling, particles were dried in-line using two silica gel diffusion driers. Despite this, negative ion mass spectra were present for only 53% of the detected particles due to the accumulation of particulate water which suppresses negative ion formation (Neubauer et al., 1997), as commonly observed for marine environments (Spencer et al., 2008).

The ATOFMS collected dual-polarity mass spectra of 496 individual particles with aerodynamic diameters of 0.2 – 1.5 μm from September 8-30, 2016. In addition to low ambient particle concentrations impacting the data collection rate, an instrumental issue with the time-of-flight mass analyzer, which has since been fixed, led to a significantly reduced particle hit rate (fraction of mass spectra collected per number of particles sized within the instrument) of less than 1%; however, laboratory tests showed that

the quality of the mass spectra collected were not affected. Individual mass spectra were analyzed using YAADA, a custom software toolkit for MATLAB (Allen, 2004). Mass spectra were clustered based on the presence and intensity of ion peaks within individual mass spectra using an ART-2a algorithm, with a vigilance factor of 0.8 and a learning rate of 0.05 for 20 iterations (Song et al., 1999). Mass spectral peaks were identified based on the most probable *m/z* considering previous laboratory and field studies (Toner et al., 2008; Pratt et al., 2011; Ault et al., 2013; Qin et al., 2012; Rehbein et al., 2011). The resulting clusters were manually combined into five groups, each representing an individual particle type (Section 3.2 and supplemental information). Due to the small sample number of particles, 100% of the measured particles were clustered either by ART-2a or manually. Despite the low number of collected mass spectra, the observed particle types are consistent with previous Arctic surface-based ATOFMS measurements by Sierau et al. (2014), who also obtained a similar number of mass spectra in part due to low particle number concentrations in the summertime Arctic boundary layer. The errors associated with number fractions for different particle types were calculated using binomial statistics.

## 3 Results and Discussion

### 3.1 Air Masses from the Arctic Ocean and Prudhoe Bay Oil Fields

The prevailing wind direction at Utqiaġvik is from the northeast across the Beaufort Sea (Searby and Hunter, 1971). Based on backward air mass trajectories (Figure 1) and wind direction (Figure S1), 70% of the days between August 21 – September 30, 2015 were influenced by the Arctic Ocean (~6 km northeast), with 10% of days influenced by the Prudhoe Bay oil fields (~300 km southeast) and 20% influenced by the town of Utqiaġvik (~5 km northwest)**.** Prudhoe Bay air masses traveled along the coast and were therefore influenced by both tundra and the Beaufort Sea, in addition to the emissions from the Prudhoe Bay oil fields. Here we discuss the influences from the two main source regions of interest, the Arctic Ocean and Prudhoe Bay, on atmospheric particle number and chemical composition.

From August 21 to September 20, 2015, the average number concentration of 13 – 746 nm particles during Arctic Ocean influenced air masses ($130 \pm 1$ particles cm$^{-3}$ with standard error of the mean) was nearly five times less than the average number concentration of Prudhoe Bay influenced air

masses ($920 \pm 4$ particles cm$^{-3}$ with standard error of the mean) (Figure 2). Aerosol number distributions for time periods classified as Arctic Ocean and Prudhoe Bay are shown in Figure S2; corresponding median and 25$^{th}$ and 75$^{th}$ percentile aerosol number distributions for both time periods are shown in Figure S3. Notably, the average particle mode diameter of $27 \pm 4$ nm during Prudhoe Bay influence was smaller than the average particle mode diameter of $76 \pm 40$ nm during Arctic Ocean influenced air masses, illustrating that the majority of the additional particles in the Prudhoe Bay air mass were less than 30 nm in diameter (Figure 2). Particles smaller than ~50 nm are often associated with combustion emissions, either from primary particles or nucleated particles within the emission plume, but can also be indicative of regional new particle formation. However, regional new particle formation would typically be followed by particle growth (Kulmala et al., 2004), which was not observed (Figure S2). Rather, this ultrafine particle mode was sustained over multiple hours (Figure S2), which also eliminates the possibility that these were from local vehicle emissions.

The condensation sink, a measure of how fast molecules will condense onto existing particles (Lehtinen et al., 2003), was calculated during the 2015 study Prudhoe Bay air mass periods using the method of Dal Maso et al. (2002). The average condensation sink was $6 \times 10^{-4}$ s$^{-1}$, over an order of magnitude lower than typically observed at mid-latitude and boreal forest sites (e.g. Jung et al., 2013; Dal Maso et al., 2002; Kulmala et al., 2001). Based on the simulations by Fierce et al. (2015), particle growth during transport for particles ~30-50 nm would take ~1-7 days, if coagulation-dominated due to limited condensable material. Particle growth was not observed during this study, suggesting that sufficient condensable material was not available for an observable change in particle diameter. Therefore, particles of this size could potentially be transported from Prudhoe Bay to Utqiaġvik during the average $21 \pm 7$ h transit time. Given the lack of primary ultrafine aerosol sources between Utqiaġvik and Prudhoe Bay, it is suggested that these particles were likely transported from Prudhoe Bay. Kolesar et al. (2017) previously observed Prudhoe Bay air masses to preferentially exhibit particle growth, compared to Arctic Ocean air masses. However, particle growth was not observed to occur within all Prudhoe Bay air masses during the summer, and particle growth events were not observed in September in Utqiaġvik (Kolesar et al., 2017).

Multi-year measurements of particle number size distributions were also compared for Arctic Ocean and Prudhoe Bay influenced air masses for the months of August and September (Figure 3). Prudhoe Bay air masses had a significantly (95% confidence interval) higher median concentration (407 particles $cm^{-3}$) compared to Arctic Ocean air masses (294 particles $cm^{-3}$), similar to the trends observed during the 2015 study. The median particle concentration within Arctic Ocean air masses is similar to the median particle number concentrations during August at Station Nord, Greenland (227 particles $cm^{-3}$, Nguyen et al., 2016) and Alert, Canada (~160 particles $cm^{-3}$; Croft et al., 2016), during September at Tiksi, Russia (222 particles $cm^{-3}$; Asmi et al., 2016), and within the range of observations onboard the Swedish icebreaker *Oden* from July – September during multiple central Arctic Ocean studies when the air masses were exposed to the open ocean (90-210 particles $cm^{-3}$; Heintzenberg et al., 2015). However, the median particle number concentration during August in Tiksi, Russia (383 particles $cm^{-3}$) is similar to the median concentration of Prudhoe Bay influenced air masses (407 particles $cm^{-3}$) even though the elevated number concentrations in Tiksi are due to biogenic influence leading to new particle formation and growth (Asmi et al., 2016).

For the multi-year measurements, the median Arctic Ocean influenced particle size distribution has three modes (10 nm, 35 nm, 118 nm), similar to observations during August at Alert, Canada (Croft et al., 2016), Station Nord, Greenland (Nguyen et al., 2016), and Ny-Ålesund, Svalbard (Tunved et al., 2013). The Prudhoe Bay air mass median size distribution also has a clear accumulation mode ~150 nm that is typical of summertime background Arctic aerosol seen in the previously mentioned studies. A two-sample Kolmogorov-Smirnov test on Prudhoe Bay and Arctic Ocean influenced distributions from both the multi-year and 2015 study concluded that the two distributions are not significantly different (p = 0.05) above 100 nm, despite chemical differences described below.

## 3.2 Single Particle Chemical Characterization

Analysis of the individual particle (0.1 – 4.0 µm) ATOFMS and CCSEM-EDX spectra resulted in the identification of five major single-particle types: sea spray aerosol (SSA), soot, organic carbon (OC), biomass burning, and mineral dust (Figure 4). Detailed descriptions of particle-type mass spectra and classifications can be found in the supplemental information. SSA internally mixed with nitrate ($NO_2^-$

[*m/z* -46] or NO$_3^-$ [*m/z* -62] using ATOFMS; N using EDX) and/or sulfate (SO$_3^-$ [*m/z* -80] using ATOFMS; S using EDX) were sub-classified as partially aged SSA (Qin et al., 2012; Gard et al., 1998) and are discussed further in Sections 3.2.1 and 3.2.2. Sulfate is identified as SO$_3^-$ [*m/z* -80] in SSA due to mass spectral interference between HSO$_4^-$ [*m/z* -97] and NaCl$_2^-$ [*m/z* -93,95,97] (Qin et al., 2012; Sultana et al., 2017). CCSEM-EDX identified a unique sulfur-rich particle type not observed by ATOFMS; this is consistent with previous ATOFMS studies, including an Arctic summer ship-based study (Sierau et al., 2014), that attributed a "missing" ATOFMS particle type to relatively pure ammonium sulfate particles that scatter visible radiation, but are not ionized by 266 nm radiation (Wenzel et al., 2003; Spencer et al., 2008). Based on CCSEM-EDX analysis (Figure 6), these sulfur particles likely comprised ~10 – 30% of the 0.13 – 1 µm particle number fraction during Arctic Ocean air mass influence, and ~10 – 20% of the 0.13 – 0.3 µm particle number fraction during Prudhoe Bay air mass influence. Accounting for these sulfur particles would reduce the reported ATOFMS fractions by ~5 – 15% for Arctic Ocean air mass influence, and ~5 – 10% for Prudhoe Bay air mass influence. Minor contributions were observed from biomass burning and mineral dust particles for Arctic Ocean (14 ± 4% and 14 ± 3%, respectively, by number) and Prudhoe Bay (10 ± 11% and 4 ± 7%, respectively, by number) influenced air masses (Figure 5). Wildfire smoke from on-going central Alaskan wildfires did not influence the site during the study based on air mass origin; therefore, biomass burning particles were likely from local residential heating or beach bonfires commonly seen around Utqiaġvik. The dirt roads and beaches near the field site are the likely source of the observed mineral dust. Both dust and biomass burning contributions were greatest when the wind was coming from Utqiaġvik.

### 3.2.1 Chemical Characterization of Aerosols during Arctic Ocean Air Mass Influence

Based on HYSPLIT backward air mass trajectories, periods of Arctic Ocean air mass influence occurred between September 8 – 12, 14 – 22 and 26 – 30. Fresh SSA contributed 80 – 100%, by number, to the measured 1 – 4 µm, as measured by CCSEM-EDX (Figure 6), consistent with previous Utqiaġvik measurements which demonstrated that SSA comprises approximately 70% of the summertime Arctic supermicron (1-10 µm) particle mass (Quinn et al., 2002). Approximately 20% of 0.13 – 0.4 µm and 40

– 70% of 0.4 – 1 μm particles, by number, were identified as fresh SSA, as determined by CCSEM-EDX, and in agreement with the measured ATOFMS number fraction of 63 ± 5% for 0.2 – 1.5 μm particles (Figure 5). Prominent chloride peaks, including $Cl^-$ [$m/z$ -35/37], $NaCl_2^-$ [$m/z$ -93/95] and $Na_2Cl^+$ [$m/z$ 81/83] (Ault et al., 2014; Gard et al., 1998), were present in the ATOFMS SSA mass spectra. The majority of the identified supermicron SSA (>99%, by number) also showed little evidence of atmospheric processing through addition of nitrogen or sulfur, identified as nitrate and sulfate by ATOFMS (Figure 4), in part due to local SSA production. Minimal chloride depletion was observed for supermicron SSA particles during Arctic Ocean influence, with an average Cl/Na mole ratio of 0.99 for 1 – 4 μm (15% depletion) (Table 1 and Figure S4), compared to the seawater Cl/Na ratio of 1.16 (Keene et al., 1986). Supermicron SSA particles also had low S/Na and N/Na mole ratios (0.15 and >0.1, respectively), indicating low contributions from sulfate and nitrate on the particles. In fact, the S/Na mole ratio of 0.15 for supermicron SSA is near the ratio expected of seawater (0.121) (Keene et al., 1986), indicating that very little atmospheric processing occurred, consistent with local SSA production. Comparatively, submicron (0.13 – 1 μm) SSA had a lower Cl/Na mole ratio (0.81, 30% depletion), as well as higher S/Na and N/Na mole ratios (0.36 and 0.27, respectively), indicating increased atmospheric processing (Williams et al., 2002; Gong et al., 2002; Laskin et al., 2002; Hopkins et al., 2008). As residence times for submicron particles are longer compared to supermicron particles, submicron SSA can be transported further, providing longer periods of atmospheric processing and leading to the observed increases in sulfate and nitrate, coincident with chloride depletion.

OC particles contributed 27%, by number, to 0.13 – 1 μm particles with minimal size dependence (Figure 6). OC contributed ~10%, by number, from 1 – 2 μm particles, with no OC particles measured between 2 and 4 μm. For the submicron OC particles, 94%, by number, were internally mixed with sulfur with an average atomic composition of 11% during Arctic Ocean influence (Table 1). Sulfur was identified as sulfate using ATOFMS spectral markers (Figure 4). The Arctic Ocean has previously been shown to be a significant source of biogenic sulfur in the form of dimethyl sulfide (DMS) (Ferek et al., 1995). DMS oxidizes in the atmosphere to form methanesulfonic acid (MSA), previously observed in Arctic aerosols (e.g. Sharma et al., 2012; Geng et al., 2010; Tjernström et al., 2014; Quinn et al., 2009; Quinn et al., 2007).

### 3.2.2 Chemical Characterization of Transported Prudhoe Bay Aerosols

For air masses influenced by Prudhoe Bay emissions, increased number fractions of soot, OC, and partially aged SSA particles were measured, with increased soot and OC expected based on previous estimates of soot (1.9 kt) and OC (2.0 kt) emissions from 2004 US Arctic oil and gas extraction activities, primarily at Prudhoe Bay (Peters et al., 2011). ATOFMS analyses identified $32 \pm 18\%$, by number, of 0.2 – 1.5 μm particles as OC (Figure 5). CCSEM-EDX identified OC particles to comprise 60%, by number, of 0.13 – 0.3 μm particles, with contributions decreasing to 10%, by number, for 0.8 – 1 μm particles and 5%, by number, for supermicron (1 – 2 μm) particles (Figure 6). ATOFMS identified hydrocarbon markers within the OC particles (e.g. $C_2H_3^+$ [$m/z$ 27], $C_3H_2^+$ [$m/z$ 37], $C_4H_2^+$ [$m/z$ 50]), similar to those detected in previous studies of vehicle combustion (Toner et al., 2008). The presence of oxidized OC was also identified ($C_2H_3O^+$[$m/z$ 43]) in these OC particles, suggesting secondary organic aerosol formation (Qin et al., 2012). However, as particle growth was not observed during Prudhoe Bay air mass influence (Section 3.1), it is likely that SOA contributions to particle mass were minor. Ammonium signal ($NH_4^+$ [$m/z$ 18]) was also detected in the OC particles. Sulfur and nitrogen were identified in 60% and 28%, by number, respectively, of OC particles between 0.13 – 1 μm, confirmed as sulfate ($HSO_4^-$ [$m/z$ -97]) and nitrate ($NO_3^-$ [$m/z$ -62]), respectively, by ATOFMS (Figure 4) (Pratt and Prather, 2009). Internally mixed sulfate and nitrate have been shown to increase the hygroscopicity of organic particles and therefore enhance their CCN activity (Petters and Kreidenweis, 2007; Wang et al., 2010).

Similar number fractions of fine mode soot particles were observed by CCSEM-EDX during both Prudhoe Bay and Arctic Ocean periods (5 – 10% and 5 – 20%, by number, across 0.13 – 1 μm, respectively) (Figure 6). Soot was also identified by ATOFMS during Prudhoe Bay periods by $C_n^+$ clusters ($C^+$ [$m/z$ 12], $C_2^+$ [$m/z$ 24], $C_3^+$ [$m/z$ 36], etc). Elevated black carbon mass concentrations (up to 0.27 μg/m$^3$) were also measured by the aethalometer during the Prudhoe Bay air mass observed on August 25 (Figure S5). Soot particles are primarily emitted through diesel combustion from heavy duty vehicles (Spencer et al., 2006) and ships (Ault et al., 2009). However, the majority of soot is expected to be less than 100 nm in diameter and therefore not chemically characterized in this study. During the 2012 ACCESS campaign off the coast of Norway, Roiger et al. (2015) observed increased soot mass concentrations <80 nm in diameter while sampling near oil and gas extraction facilities, consistent with

the observed elevated ultrafine particle number concentrations in the present study when under Prudhoe Bay air mass influence (Figure 2 and 3).

Since the air mass trajectory from Prudhoe Bay to Utqiaġvik crosses the Beaufort Sea, SSA particles were still a major contributor, making up over 90% of supermicron (1 – 4 μm) particles by number. However, unlike the Arctic Ocean air mass influence, ~60%, by number, of the supermicron SSA was classified as partially aged SSA. This is over three times the fraction compared to Arctic Ocean air masses (16%) due to atmospheric processing during $21 \pm 7$ hour transit over land before reaching Utqiaġvik. SSA shows 43% chloride depletion in the SSA EDX spectra (Cl/Na mole ratio of 0.66 compared to 1.16 in seawater (Keene et al., 1986)) (Table 1); ATOFMS chloride peak intensities ($NaCl_2^-$ [$m/z$ -93/95], $Cl^-$ [$m/z$ -35/37]) are lower than during the Arctic Ocean influence for the SSA particles. Sulfur, identified as sulfate ($SO_3^-$ [$m/z$ -80]) in ATOFMS spectra (Pratt and Prather, 2009), was internally mixed with 86%, by number, of SSA (Table 1). For these particles, the S/Na mole ratio of the submicron (0.13 – 1 μm) SSA during Prudhoe Bay influence (0.53) is higher than expected from seawater (0.121), indicating contributions of secondary sulfate (Keene et al., 1986). Nitrogen, identified as nitrate ($NO_2^-$ [$m/z$ -46] and $NO_3^-$ [$m/z$ -62]) by ATOFMS (Liu et al., 2003), was observed in 40%, by number, of the SSA particles by CCSEM-EDX. Similar to S/Na mole ratios, submicron (0.13 – 1 μm) SSA N/Na ratios were substantially higher during Prudhoe Bay influence (0.54) compared to Arctic Ocean influenced SSA (0.27). In addition to longer atmospheric residence time for submicron particles leading to increased submicron atmospheric processing (Williams et al., 2002; Gong et al., 2002), models have found that secondary species such as sulfate and nitrate preferentially accumulate on submicron particles (Bassett and Seinfeld, 1984). These SSA particles were likely transported from the Arctic Ocean surrounding Prudhoe Bay, and underwent chloride displacement during transport due to multiphase reactions with N- and S- containing trace gases from precursor Prudhoe Bay combustion emissions ($SO_2$ and $NO_x$) leading to nitrate (Hara et al., 1999) and sulfate (Hara et al., 2003) formation. A previous ATOFMS study during the summertime in the high Arctic Ocean detected similar partially aged SSA particles containing nitrate and sulfate with low intensity chloride markers (Sierau et al., 2014).

## 4 Conclusions

The chemical composition of individual atmospheric particles transported to Utqiaġvik, Alaska from the Arctic Ocean and Prudhoe Bay were measured from August 21 to September 30, 2015. During periods of Arctic Ocean influence, fresh SSA was the major contributor to both submicron (~20%, by number, from 0.13 – 0.4 µm, 40 – 70% between 0.4 – 1 µm) and supermicron (80 – 100%, by number, from 1-4 µm) particles with only 30% chloride depletion (average Cl/Na mole ratio of 0.81) for all submicron SSA and 15% chloride depletion (average Cl/Na mole ratio of 0.99) for all supermicron SSA. Submicron OC particles contributed an average of 27%, by number, from 0.13 – 1 µm with a minimum of 10%, by number, from 0.13 – 0.2 µm and were likely from a marine biogenic source. With complete summertime sea ice loss expected by 2050 (Wang and Overland, 2015; Overland and Wang, 2013), increasing aerosol and trace gas emissions from the open Arctic Ocean are expected (Browse et al., 2013; Struthers et al., 2011).

Increased total particle number concentrations ($920 \pm 4$ particles cm$^{-3}$) and a smaller particle size mode of $27 \pm 4$ nm were observed during periods of Prudhoe Bay air mass influence, in comparison to Arctic Ocean air masses ($130 \pm 1$ particles cm$^{-3}$, $76 \pm 40$ nm, respectively), due to transportation of ultrafine combustion particles from the Prudhoe Bay oil fields. These transported particles have the potential to grow (Kolesar et al., 2017)  and serve as CCN, which would have a large impact on the low CCN concentrations currently in the Arctic (Mauritsen et al., 2011). During these periods, increased number fractions of partially aged SSA ($28 \pm 1\%$, by number, of particles 0.13 – 4 µm) and OC (60%, by number, of 0.13 – 0.3 µm particles with a minimum of 10%, by number, of 0.8 – 1 µm particles) were observed by CCSEM-EDX, with evidence of sulfate and nitrate internally mixed with SSA and OC particles due to heterogeneous reactions and gas-particle partitioning, respectively, during transport. Increased particle aging has been shown previously to increase the CCN activity of combustion particles (Furutani et al., 2008; Petzold et al., 2005). Therefore, increasing trace gas and aerosol emissions due to Arctic oil and gas extraction activities will contribute to further Arctic climate change (Law and Stohl, 2007).

*Competing Interests.* The authors declare that they have no competing financial interests.

*Acknowledgements*. This study was supported by the NOAA Climate Program Office Atmospheric Chemistry, Carbon Cycle, and Climate Program, through NA14OAR4310149 (University of Michigan) and NA14OAR4310150 (Baylor University). Funding for housing and logistical support was provided by Department of Energy Atmospheric Radiation Measurements (DOE ARM) field campaign 2013-6660.

UIC-Science and Department of Energy Atmospheric Radiation Measurement Climate Research Facility are thanked for logistics assistance in Utqiaġvik, AK. CCSEM-EDX analyses were performed at the Environmental Molecular Sciences Laboratory (EMSL), a national scientific user facility located at the Pacific Northwest National Laboratory (PNNL) and sponsored by the Office of Biological and Environmental Research of the U.S DOE. PNNL is operated for DOE by Battelle Memorial Institute

under Contract No. DE-AC06-76RL0 1830. Travel funds to PNNL were provided by the University of Michigan Rackham Graduate School. Additional CCSEM-EDX analyses were carried out at the Michigan Center for Materials Characterization. Andrew Ault (University of Michigan) is thanked for discussions of aerosol sampling and CCSEM-EDX analysis. We also thank the National Oceanic and Atmospheric Administration (NOAA) Global Monitoring Division (including Anne Jefferson, staff at the Barrow

Observatory, and Wolfram Birmili (Leibniz Institute for Tropospheric Research) for meteorological and long-term aerosol data.

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

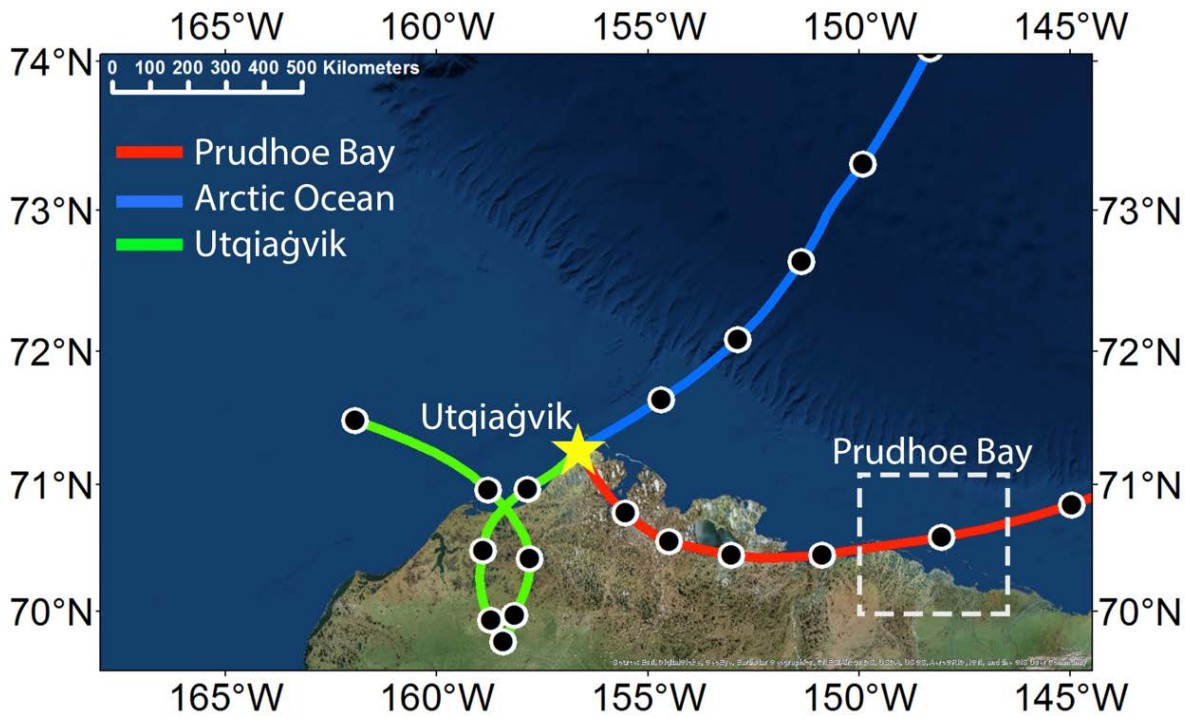

**Figure 1.** Average 48 h HYSPLIT backward air mass trajectories for three major areas of influence: Prudhoe Bay, the ice-free Arctic Ocean, and the town of Utqiaġvik. 6 h time intervals are indicated by black circles. The Utqiaġvik, AK sampling site is indicated by the yellow star, and the area of the greatest Prudhoe Bay emissions influence is indicated by the white dashed square as defined by Kolesar et al. (2017). The map background was provided by ArcGIS 10.3.1 with the World Imagery basemap (Sources: Esri, DigitalGlobe, Earthstar Geographics, CNES/Airbus DS, GeoEye, USDA FSA, USGS, Getmapping, Aerogrid, IGN, IGP, and the GIS User Community).

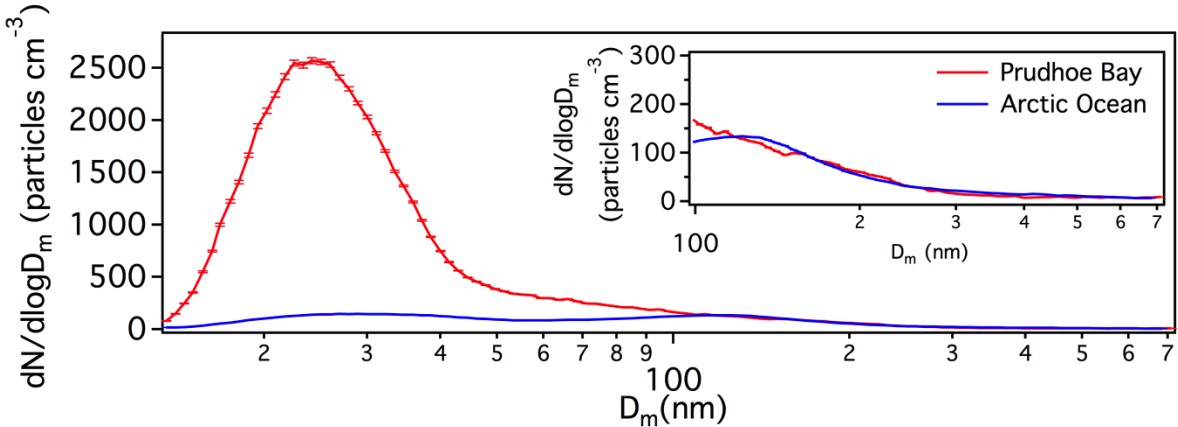

**Figure 2.** Average, and standard error of the mean, particle number size (mobility diameter) distributions during Prudhoe Bay and Arctic Ocean influenced air masses from August 21–September 20, 2015, with the above 100 nm distributions inset. The full time series of the time-varying aerosol distribution is shown in Figure S2.

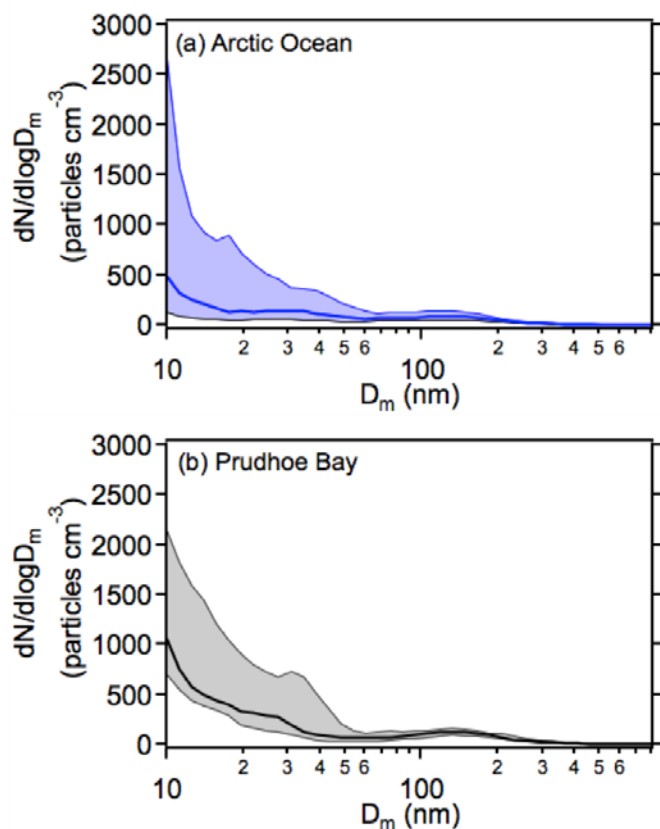

**Figure 3.** Particle number distribution for (a) Arctic Ocean and (b) Prudhoe Bay air masses observed for August-September 2008, 2009, 2013, and 2014 (median shown by the solid line, 25[th] and 75[th] percentiles shaded) at the NOAA Barrow Observatory.

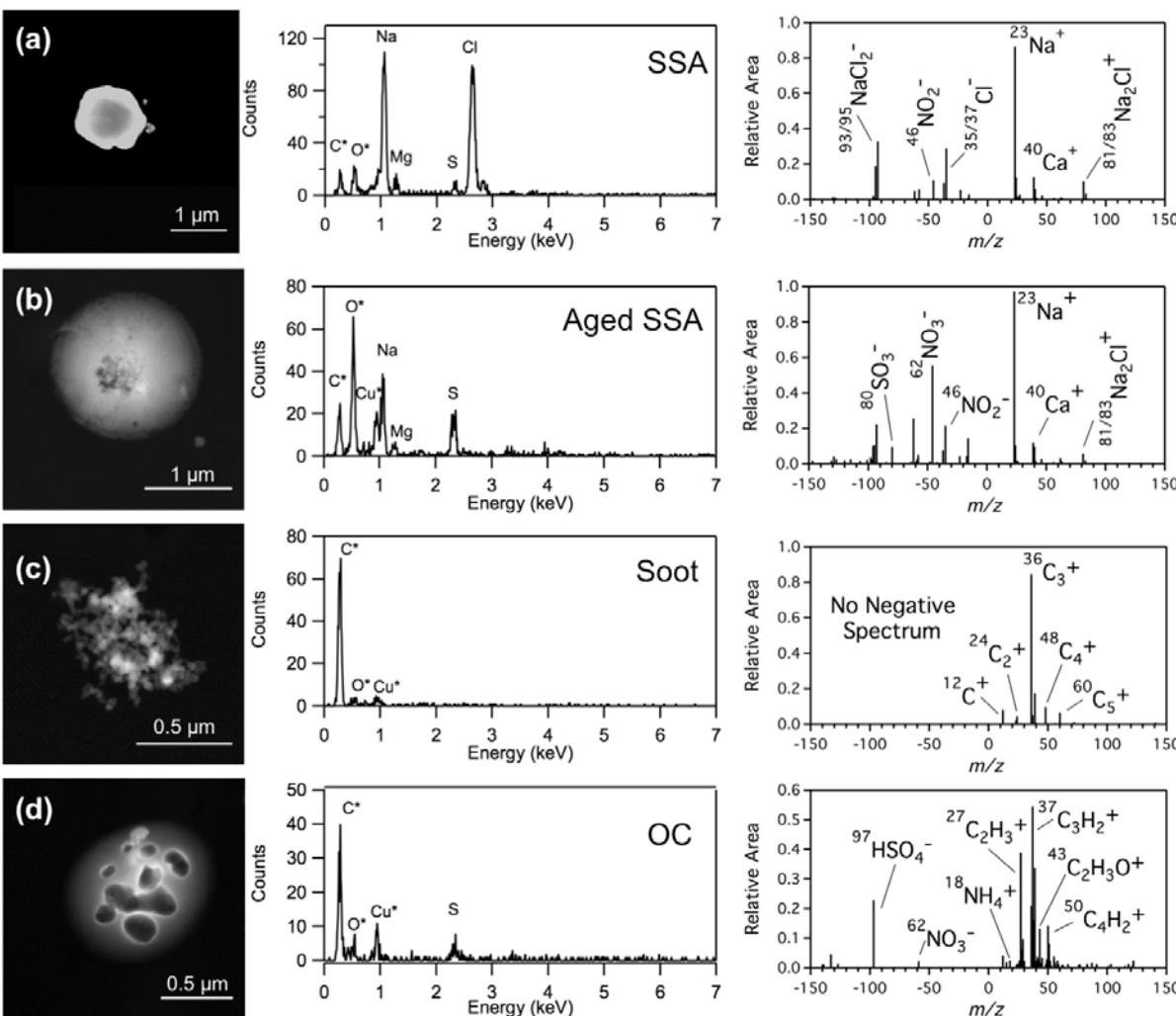

**Figure 4.** Representative SEM images (left) and EDX spectra (middle), as well as average ATOFMS mass spectra (right), for the major particle types observed: (a) Sea Spray Aerosol (SSA), (b) Partially Aged SSA, (c) Soot, (d) Organic Carbon (OC).

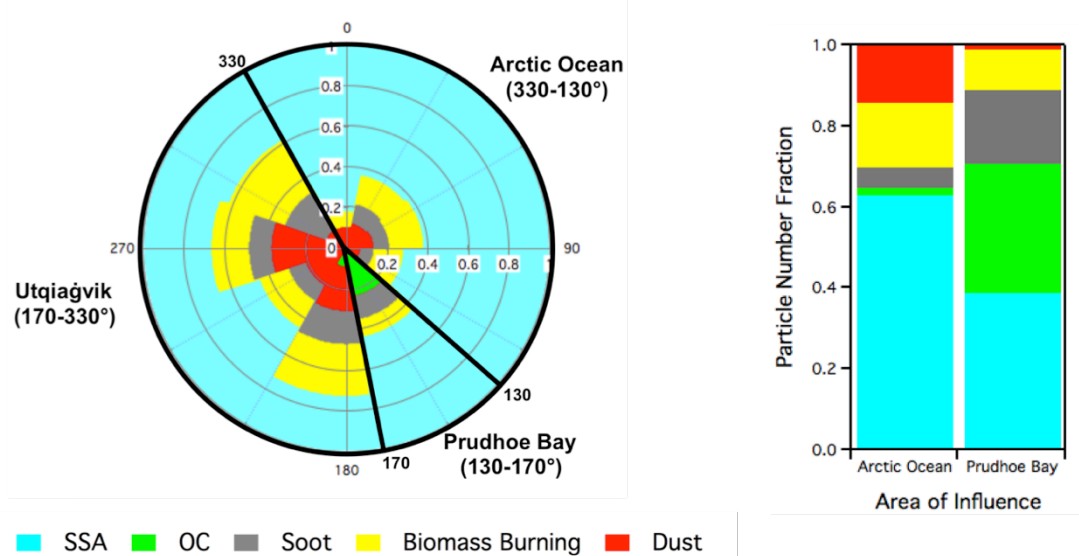

**Figure 5.** ATOFMS individual particle composition (0.2-1.5 μm) number fractions for 496 analyzed particles from September 8-20, 2015, based on wind direction (left) and air mass influence (right), determined by backward air mass trajectories. Data were binned every 40 degrees.

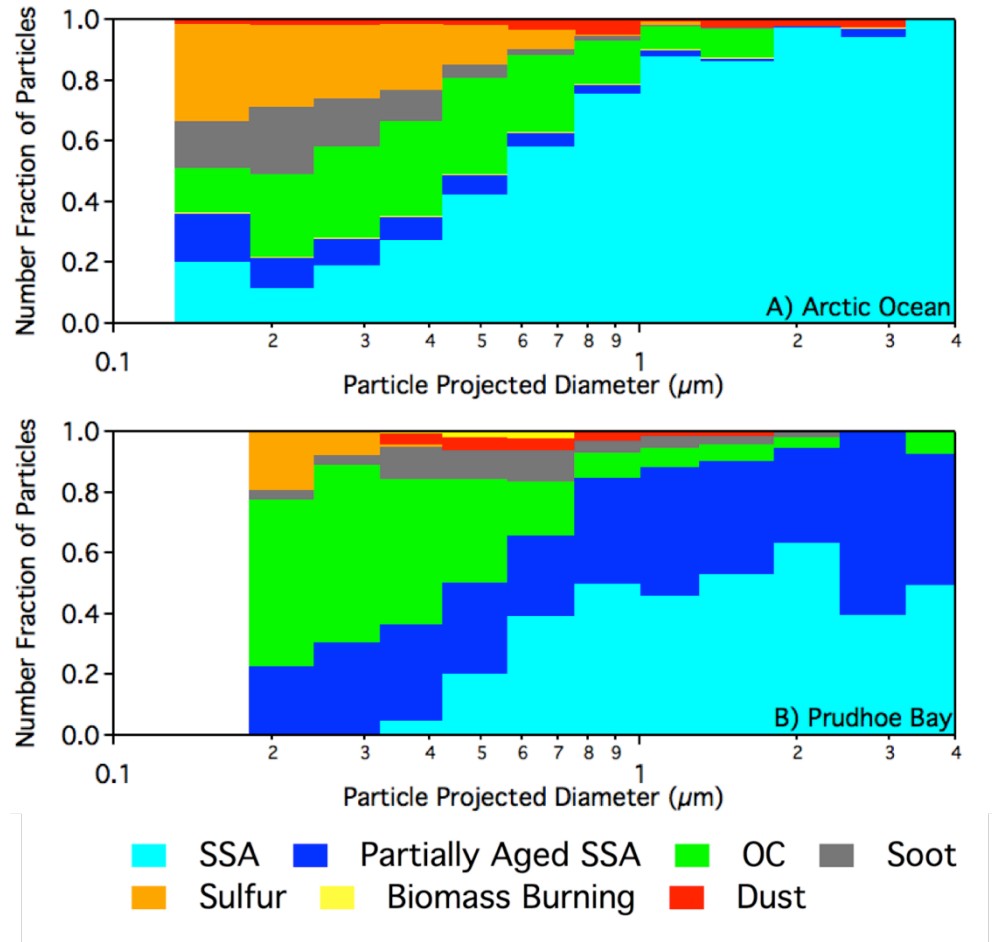

**Figure 6.** Size and chemical composition of individual particles measured by CCSEM-EDX during influence by (A) Arctic Ocean (2,869 particles analyzed) and (B) Prudhoe Bay (1,997 particles analyzed) air masses. For Arctic Ocean influenced periods, the following 8 h samples were analyzed: September 8, 2015 (00:00–08:00, 08:00 – 16:00), September 9, 2015 (00:00–08:00), September 15, 2015 (00:00–08:00). For Prudhoe Bay influenced periods, the following 8 h samples were analyzed: September 23, 2015 (00:00–08:00, 08:00–16:00). All times are in AKDT.

**Table 1.** Submicron and supermicron CCSEM-EDX number fractions and mole ratios for sulfate, nitrate, and chloride within individual SSA particles (SSA and partially aged SSA classes combined) during Arctic Ocean and Prudhoe Bay air masses. S, N and Cl were confirmed as sulfate, nitrate, and chloride by ATOFMS.

| SSA Projected Area Diameter | Sulfate (Number Fraction) | Nitrate (Number Fraction) | Chloride (Number Fraction) | S/Na | N/Na | Cl/Na |
|---|---|---|---|---|---|---|
| **Arctic Ocean** | | | | | | |
| 0.13 – 1 μm | 0.77 | 0.33 | 0.81 | 0.36 | 0.27 | 0.81 |
| 1 – 4 μm | 0.97 | 0.22 | 0.99 | 0.15 | <0.1 | 0.99 |
| **Prudhoe Bay** | | | | | | |
| 0.13 – 1 μm | 0.86 | 0.40 | 0.87 | 0.53 | 0.54 | 0.67 |
| 1 – 4 μm | 0.90 | 0.43 | 0.87 | 0.32 | 0.76 | 0.55 |