# Peer review of "Contributions of Transported Prudhoe Bay Oil Field Emissions to the Aerosol Population in Utqiagvik, Alaska"

_Atmospheric Chemistry and Physics, 2017_

## Referee Comment (RC1) · Anonymous Referee #1 · 5 Jun 2017

This manuscript describes single particle characterization (ATOFMS and CCSEM-EDX) of particulate matter attributed to Arctic oil extraction activities at Prudhoe Bay, Alaska, and contrasts this with particulate matter dominated by natural emissions from the nearby Arctic Ocean for ∼1 month of measurements in late summer. This work contributes to the very few studies of local Arctic emissions of anthropogenic aerosol, and is therefore valuable in extending our understanding of local Arctic pollution sources in the context of the summertime Arctic natural background. The manuscript is overall well written and clear. It merits publication in ACP after the following comments have been addressed.

Major Comments:

In the introduction as well as in the discussion sections, the authors should make further efforts to ensure that proper, complete, and most importantly accurate, credit is given to prior related work. Specific instances are indicated in the minor comments.

Minor Comments:

Introduction: This introduction gives no context for the conditions expected in the unpolluted summer Arctic. The seasonal cycle in Arctic aerosol is very relevant to the context of these measurements, especially under the cleaner conditions of summer to autumn. Similarly, an acknowledgment of pollution influences in winter and spring is also relevant. The natural question here is how Prudhoe Bay emissions might compare to influences of long range transport in terms of aerosol loading and composition.

P3 L10-15: The discussion of BC and OC contributions from total Arctic oil and gas extraction, versus that from Prudhoe Bay is at present somewhat confusing. In addition, the methods used to arrive at BC and OC estimates might be relevant to this discussion, e.g., were in-situ measurements used to arrive at these estimates?

P3 L17: Are these US contributions from Prudhoe Bay alone or does this estimate represent expansion to other sites?

P3 L20: It is not strictly true to suggest that emission of gas phase species would lead to solely new particle formation. If the intention is to highlight the role that the very low condensation sinks of the summer Arctic could play in this respect, then the authors should state this clearly.

P3 L22: Flanner et al., 2013 does not indicate anything about the impact of BC on clouds. In abstract the authors state that the study has limitations owing to their lack of treatment of indirect effects.

P3 L25: Citation of other relevant work, such as Flanner et al, 2007 (and others) is advisable here

[Figure]

P3 L27: Is this result directly from Koch 2009, or is it elaborated by Bond 2013?

P3 L27-28: Did the modelling studies discussed here include local Arctic BC sources? This is worth discussing.

P4 L4: To do justice to the study of Barrett 2015 the authors should discuss the varying contribution of fossil fuels to Barrow EC over the winter period. Biomass burning is also an important source and can have local Arctic sources.

P4 L9: Do the authors suggest that due to the focus on biomass burning, the results of this study are less relevant to their discussion? The measurements of Brock 2011 likely represent some of the few more complete chemical characterizations of Prudhoe Bay aerosol emissions.

P4 L22: Cappa 2012 explicitly shows that absorption enhancement was not detectable, despite significant particle mixing, during their measurement campaign. This needs to be much more clearly stated, and the complexity of this issue need to be addressed. In particular, I do not agree with the statement that light absorption is enhanced the presence of sulfate or organic species in the same particles. Some studies have observed absorption enhancements (e.g., Knox 2009, Liu 2015) while others have not (e.g., Cappa 2012, Healy 2015). It is not clear whether this effect is significant in all ambient aerosol and this issue should not be stated as being entirely clear.

P4 L26: Is such a binary definition relevant to ambient particles? It might be more accurate to state that ambient aerosol can be found with range of mixing states approaching external or internal mixing in different cases and with influence from different source types.

Method section: Please ensure that all manufacturer information is as complete as possible with relevant names, models and locations.

P5 L21: How do back trajectories compare for the two sites?

P6 L23: Given the long sampling period and the acknowledgement of local vehicle

emission contributing to particle measurements, how might fast-varying local emissions (such as vehicles or generators) influence the particle composition interpreted from these 8h measurements?

Section 2.3: How is statistical significance determined for these measurements? What is the minimum number of particles that must be analyzed in order to to have a representative sample?

P8 L10: This is a remarkably small number of particle spectra to draw conclusions from. While I acknowledge the challenges of obtaining a large number of particle spectra in such a low concentration environment, the authors should acknowledge this limitation of the ATOFMS measurement in a more prominent manner (for example, Sierau 2014 acknowledges this challenge in their abstract). Related to this, what fraction of ATOFMS detected particles did not provide mass spectra? These limitations in no way contradict the main conclusions of the work, since the CCSEM-EDX analysis provides a complementary data set that provides similar conclusions; however, this limitation needs to be clearly stated.

P9 up to L25: Emission of small, primary particles from Prudhoe Bay emissions is likely also associated with emission of oxidizable or condensible gas phase species. Why might the Prudhoe emissions have stayed relatively small during transport to your measurement site?

P10 L8: Is comparison to Alert, Nunavut also possible?

P10 L20: However, your measurements show that the chemical composition of particles >100nm is different between the two types of air masses, and I doubt that no influence on these particle sizes occurs.

P11 L1: This work is from Summit, Greenland, which is arguably quite different from other, lower altitude, regions of the Arctic for a few reasons. First, ascent over Greenland can cause deposition of transported species leading to quite clean air masses.

Second, the lack of local sources (aside from snow pack photochemistry) compared to other Arctic regions which are directly subject to marine, coastal, and tundra influences. While some recent studies have suggested the presence of condensible material in the summer Arctic (e.g., Mungall 2017, Willis 2016), it would be very surprising if Prudhoe Bay did not also emit gas phase species. The authors might want to discuss what is known about gas phase emissions from oil and gas extraction. Some studies from lower latitudes (e.g., Liggio 2016 ) suggest these can be substantial.

P11 L5: A contradiction of your observations by those of Kolesar 2017 does not help to make your argument clearer here. If particle growth from Prudhoe Bay was observed previously what differences do we expect in late summer 2015? Was the time of year significant (Kolesar 2017 shows growth peaking in June to July)? What the meteorology different? This really needs further discussion.

P11 L20: Given that the ATOFMS misses sulfur-rich particles, which should be a reasonable contributor to Arctic Ocean air masses owing to DMS oxidation, how reliable are ATOFMS estimates of the fractional particle number contribution from various particle types? Is there some was to estimate the fraction of particles that are detected by the ATOFMS but not desorbed and ionized due to being sulfate rich? Table 1 would suggest a large fraction of particles contained sulfate. What fraction of particles scattered light in the ATOFMS, but did not produce mass spectra?

P11 L23: Would these sources not have been captured during the 'local' or Utqiagvik influenced periods?

P12 section 3.2.1: Besides trajectories and wind-sector analysis, what chemical characteristics do the authors have for the Arctic Ocean sector being representative of a clean marine background for the local region? The presence of BC-containing particles as well as aged SSA showing nitrate peaks (suggesting NOx chemistry), while I acknowledge that sulfate peaks could arise from interaction with DMS oxidation products, suggest a pollution influence (at least to some extent) on these air masses as

well. Do these air masses, either at the measurement site or at the Barrow Observatory, conform the the thresholds in BC (or absorbing aerosol) usually associated with clean marine conditions (e.g., < 50 ng/m3 of BC (Gantt 2013))?

P13 L10: Do these estimates still correspond to 2004, or rather a yearly average?

P13 L17: Is this single fragment really an unambiguous indicator of SOA formation? If SOA formation was occurring on Prudhoe bay emissions, why did the particles remain quite small, or put another way why do the authors suggest earlier in the text that particle growth did not occur?

P13 L27: It is difficult to draw conclusions from these differences, especially when the number of detected particles is so small. Do long term aerosol absorption data from the Barrow Observatory help with these conclusions at all?

P14 L16: Why is the main sulfate peak used to identify sulfur specifc different here compared to line 20 on the previous page?

P14 L20: Does a peak at NO2- indicate nitrite or just fragmentation of nitrate?

Figure 1: Average trajectories for the study period might be more meaningful that selected trajectories. Additionally, is the area North of Utqiagvik completely ice free during the measurement, as pictured?

Figure 2: Percentiles to illustrate the range of the data might be more appropriate here

Figure 4: Were elemental carbon peaks observed in OC particles? And similarly, was OC present on the EC particles detected? Can carbon and oxygen peaks be quantified in the CCSEM-EDX spectra?

Figure 5: The number of detected particles should be noted in this figure caption

Supplement: Is mention of the TMA containing particles warranted in the main body of the paper? It helps to show different source influences, since these were presumably detected during Arctic Ocean influence. Or, are there too few particle spectra with TMA

peaks to draw conclusions?

Specific comments:

P3 L4: natural gas P3 L7: Which types of pollutants are relevant here? Maybe list the major ones we expect, that are then discussed in following paragraphs? P4 L13: "primary aerosol can. . ." P4 L22: What is meant by "mostly" here? P9 L17: perhaps: "Arctic Ocean influenced periods" P9 L19: Prudhoe Bay air masses P15 L26: respectively P16 L5: "may contribute to further decrease"

References:

Flanner, M. G.: Arctic climate sensitivity to local black carbon, J. Geophys. Res-Atmos., 118, 1840-1851, 2013.

Flanner, M. G., Zender, C. S., Randerson, J. T., & Rasch, P. J. Present-day forcing and response from black carbon in snow. J. Geophys. Res. 112, doi:10.1029/2006JD008003, 2007.

Gantt, B. and Meskhidze, N.: The physical and chemical characteristics of marine primary organic aerosol: a review, Atmos. Chem. Phys., 13, 3979-3996, doi:10.5194/acp-13-3979-2013, 2013.

Healy, R. M., J. M. Wang, C.-H. Jeong, A. K. Y. Lee, M. D. Willis, E. Jaroudi, N. Zimmerman, N. Hilker, M. Murphy, S. Eckhardt, A. Stohl, J. P. D. Abbatt, J. C. Wenger, and G. J. Evans (2015), Light-absorbing properties of ambient black carbon and brown carbon from fossil fuel and biomass burning sources. J. Geophys. Res. Atmos., 120, 6619–6633. doi: 10.1002/2015JD023382.

Knox, A., Evans, G. J., Brook, J. R., Yao, X., Jeong, C. H., Godri, K. J., Sabaliauskas, K., and Slowik, J. G.: Mass absorption cross-section of ambient black carbon aerosol in relation to chemical age, Aerosol Sci. Tech., 43, 522–532, 2009.

Liggio, J et al., Oil sands operations as a large source of secondary organic aerosols,

Nature, doi:10.1038/nature17646, 2016

Liu, S., Aiken, A. C., Gorkowski, K., Dubey, M., Cappa, C. D., Williams, L. R., Herndon, S. C., Massoli, P., Fortner, E. C., Chhabra, P., Brooks,W., Onasch, T. B., Jayne, J. T.,Worsnop, D., China, S., Sharma, N., Mazzoleni, C., Xu, L., Ng, N., Liu, D., Allan, J. D. Lee, J., Fleming, Z. L. Mohr, C., Zotter, P., Szidat, P., and Prevot, A. S. H.: Enhanced light absorption by mixed source black and brown carbon particles in UK winter, Nature Communications, 6, 8435, doi:10.1038/ncomms9435, 2015.

Mungall, E et al., Microlayer source of oxygenated volatile organic compounds in the summertime marine Arctic boundary layer, PNAS, doi: 10.1073/pnas.1620571114, 2017.

Willis, M. D., Burkart, J., Thomas, J. L., Köllner, F., Schneider, J., Bozem, H., Hoor, P. M., Aliabadi, A. A., Schulz, H., Herber, A. B., Leaitch, W. R., and Abbatt, J. P. D.: Growth of nucleation mode particles in the summertime Arctic: a case study, Atmos. Chem. Phys., 16, 7663-7679, doi:10.5194/acp-16-7663-2016, 2016.

---

## Referee Comment (RC2) · Anonymous Referee #2 · 4 Jul 2017

General comments:

In this study, the authors present observations of particle composition and size distributions from the Barrow Environmental Observatory. The authors compare and contrast observations of aerosol from air masses that originated over the Arctic Ocean with aerosol from air masses that originated from the direction of the Prudhoe Bay oil fields.

I recommend the manuscript for publication, provided that my following points, and those of the first reviewer, are addressed. In order to reduce redundancy for the authors' response, I will restrict my points to those not covered by the first reviewer.

Specific comments:

[Figure]

P3 Lines 14-16: Stohl et al. (2013) makes no attribution of PM or OC from Prudhoe Bay.

P9 Lines 18-25: I think perhaps that the authors intend to exclude regional-scale nucleation events here, but not nucleation within emissions plumes from the Prudhoe Bay oil fields. The authors even suggest that the emissions from such drilling operations can contribute to new-particle formation on page 3, lines 19-21. Observations of particle growth would be expected for a regional-scale nucleation event, but not for continuous nucleation within an emitted plume of condensable gases, such as those observed by Brock et al. (2002) and Brock et al. (2003). If there was continuous new-particle formation occurring within a plume from the Prudhoe Bay oil field, under steady-state conditions (including constant wind speed and direction), no change in the size of observed aerosol would be observed unless the distance from the source changed. The age of the aerosol between nucleation in the plume and detection at the observation site would be constant with time, and therefore so would the size of the aerosol.

Technical corrections:

P11 Line 16: "(Sierau et al. (2014)" should be "(Sierau et al., 2014)"

Citations:

Brock, C. A., Washenfelder, R. A., Trainer, M., Ryerson, T. B., Wilson, J. C., Reeves, J. M., Huey, L. G., Holloway, J. S., Parrish, D. D., Hübler, G. and Fehsenfeld, F. C.: Particle growth in the plumes of coal-fired power plants, J. Geophys. Res., 107(D12), doi:10.1029/2001JD001062, 2002.

Brock, C. a., Trainer, M., Ryerson, T. B., Neuman, J. A., Parrish, D. D., Holloway, J. S., Nicks, D. K., Frost, G. J., Hübler, G., Fehsenfeld, F. C., Wilson, J. C., Reeves, J. M., Lafleur, B. G., Hilbert, H., Atlas, E. L., Donnelly, S. G., Schauffler, S. M., Stroud, V. R. and Wiedinmyer, C.: Particle growth in urban and industrial plumes in Texas, J. Geophys. Res. Atmos., 108(D3), doi:10.1029/2002JD002746, 2003.

Stohl, A., Klimont, Z., Eckhardt, S., Kupiainen, K., Shevchenko, V., Kopeikin, V., and Novigatsky, A.: Black carbon in the Arctic: the underestimated role of gas flaring and residential combustion emissions, Atmos. Chem. Phys., 13, 8833-8855, 2013.

---

## Author Comment (AC1) · 29 Jul 2017

**Response to Reviewer #1**

**We thank the reviewer for their helpful comments and suggestions. We provide below the original comments, shown in gray, and our responses, with specific revisions noted, in bold font.**

This manuscript describes single particle characterization (ATOFMS and CCSEM- EDX) of particulate matter attributed to Arctic oil extraction activities at Prudhoe Bay, Alaska, and contrasts this with particulate matter dominated by natural emissions from the nearby Arctic Ocean for ~1 month of measurements in late summer. This work con- tributes to the very few studies of local Arctic emissions of anthropogenic aerosol, and is therefore valuable in extending our understanding of local Arctic pollution sources in the context of the summertime Arctic natural background. The manuscript is overall well written and clear. It merits publication in ACP after the following comments have been addressed.

Major Comments:

In the introduction as well as in the discussion sections, the authors should make further efforts to ensure that proper, complete, and most importantly accurate, credit is given to prior related work. Specific instances are indicated in the minor comments.

**We added and revised references used throughout, with specific instances requested by the reviewer addressed in the responses to the comments below.**

Minor Comments:

Introduction: This introduction gives no context for the conditions expected in the unpolluted summer Arctic. The seasonal cycle in Arctic aerosol is very relevant to the context of these measurements, especially under the cleaner conditions of summer to autumn. Similarly, an acknowledgment of pollution influences in winter and spring is also relevant. The natural question here is how Prudhoe Bay emissions might compare to influences of long range transport in terms of aerosol loading and composition.

**We added a statement to the introduction (P3 L9-13) addressing pollution contributions from oil and gas extraction in regards to the seasonal cycle, which states: "The Arctic aerosol population is characterized by a maximum mass loading in the winter, due to transported pollutants from the mid-latitudes, and a minimum in the summer, when local sources, including sea spray aerosol, dominate (Quinn et al., 2002; Quinn et al., 2007). However, there is limited knowledge of aerosols produced within the Arctic, particularly in the context of changing emissions from both natural and anthropogenic sources (Arnold et al., 2016)."**

P3 L10-15: The discussion of BC and OC contributions from total Arctic oil and gas extraction, versus that from Prudhoe Bay is at present somewhat confusing. In addition, the methods used to arrive at BC and OC estimates might be relevant to this discussion, e.g., were in-situ measurements used to arrive at these estimates?

**We revised this paragraph to reduce confusion about what is from the overall Arctic vs. Prudhoe Bay.**

P3 L17: Are these US contributions from Prudhoe Bay alone or does this estimate represent expansion to other sites?

**These particulate emissions represent contributions from all US Arctic oil and gas extraction sites. We clarified in this paragraph that Prudhoe Bay is the primary, but not only, contributor to the estimated US oil field emissions in the Arctic.**

P3 L20: It is not strictly true to suggest that emission of gas phase species would lead to solely new particle formation. If the intention is to highlight the role that the very low condensation sinks of the summer Arctic could play in this respect, then the authors should state this clearly.

**We revised this to state "…drilling operations can emit aerosol precursors ($NO_x$, $SO_2$, and VOCs) and alter oxidant levels, which can lead to the formation of secondary aerosol, as well as contribute to new particle formation…".**

P3 L22: Flanner et al., 2013 does not indicate anything about the impact of BC on clouds. In abstract the authors state that the study has limitations owing to their lack of treatment of indirect effects.

**This was not clearly worded, and we clarified this to now state: "BC is estimated to have a warming effect on the Arctic atmosphere (e.g. Bond et al., 2013; Flanner, 2013; Flanner et al., 2007; Sand et al., 2013; Sharma et al., 2013)."**

P3 L25: Citation of other relevant work, such as Flanner et al, 2007 (and others) is advisable here

**References to Flanner et al. (2007), Sand et al. (2013), and Sharma et al. (2013) have been added to this paragraph.**

P3 L27: Is this result directly from Koch 2009, or is it elaborated by Bond 2013?

**This is directly from Koch et al. (2009), and we updated the sentence to accurately reflect this.**

P3 L27-28: Did the modelling studies discussed here include local Arctic BC sources? This is worth discussing.

**The following sentence was added on P4 L4-7 to discuss that these modeling inaccuracies may be improved by the inclusion of local Arctic BC sources. "Koch et al. (2009) suggest that Arctic BC concentrations are under-predicted by a variety of models by an average factor of 2.5, which may be improved by more accurately incorporating local BC sources (Flanner, 2013)."**

P4 L4: To do justice to the study of Barrett 2015 the authors should discuss the varying contribution of fossil fuels to Barrow EC over the winter period. Biomass burning is also an important source and can have local Arctic sources.

**Following the reviewer's suggestion, we expanded the discussion of Barrett et al. (2015) (P4 L9-11) to now state: "…similar to the recent results of Barrett et al. (2015) at Utqiaġvik during December 2012 – March 2013 when transported particles from Russian oil fields were observed. Barrett et al. (2015) also measured regional Arctic BC from both fossil fuel combustion and biomass burning."**

P4 L9: Do the authors suggest that due to the focus on biomass burning, the results of this study are less relevant to their discussion? The measurements of Brock 2011 likely represent some of the few more complete chemical characterizations of Prudhoe Bay aerosol emissions.

**We removed this statement; it was not our intention to appear to be overlooking the characterization efforts of Brock et al. (2011) in terms of the Prudhoe Bay aerosol.**

P4 L22: Cappa 2012 explicitly shows that absorption enhancement was not detectable, despite significant particle mixing, during their measurement campaign. This needs to be much more clearly stated, and the complexity of this issue need to be addressed. In particular, I do not agree with the statement that light absorption is enhanced the presence of sulfate or organic species in the same particles. Some studies have observed absorption enhancements (e.g., Knox 2009, Liu 2015) while others have not (e.g., Cappa 2012, Healy 2015). It is not clear whether this effect is significant in all ambient aerosol and this issue should not be stated as being entirely clear.

**We acknowledge that this is a complex topic and updated P4 L26-28 accordingly: "It is currently not clear whether light absorption by a BC particle is enhanced by sulfate or organic coatings (e.g., Cappa et al., 2012; Chung and Seinfeld, 2005; Healy et al., 2015; Jacobson, 2001; Knox et al., 2009; Liu et al., 2015; Moffet and Prather, 2009)."**

P4 L26: Is such a binary definition relevant to ambient particles? It might be more accurate to state that ambient aerosol can be found with range of mixing states approaching external or internal mixing in different cases and with influence from different source types.

**We clarified this on P5 L3-5, which now reads: "Ambient aerosol populations typically vary between internal mixtures, with multiple chemical species contained within a single particle, and external mixtures, with chemical species present as separate particles (Prather et al., 2008)."**

Method section: Please ensure that all manufacturer information is as complete as possible with relevant names, models and locations.

**This information has been added.**

P5 L21: How do back trajectories compare for the two sites?

**These sites are separated by 5.5 km, less than the resolution of HYSPLIT (1 degree, ~100 km), with only flat tundra located between them (P6 L1). Therefore, use of HYSPLIT to differentiate between the sites is not possible, and trajectories at the two sites are expected to be similar.**

P6 L23: Given the long sampling period and the acknowledgement of local vehicle emission contributing to particle measurements, how might fast-varying local emissions (such as vehicles or generators) influence the particle composition interpreted from these 8h measurements?

**Short particle spikes due to local emissions would have been detected as brief increases in particle number concentrations by the SMPS, operated with 5 min resolution (Figure S2). These events were only observed during the Utqiagvik influence sampling periods (e.g. Sept 7, see Figure S2); these local contributions were not observed during the clean Arctic or Prudhoe Bay influenced periods (Figure S2).**

Section 2.3: How is statistical significance determined for these measurements? What is the minimum number of particles that must be analyzed in order to to have a representative sample?

**Significance was determined using binomial statistics, which we have clarified on P8 L5-6. The minimum number of particles for a representative sample are between ~300 and 1,000. See Willis et al. (2002), Section 4.6.2.**

P8 L10: This is a remarkably small number of particle spectra to draw conclusions from. While I acknowledge the challenges of obtaining a large number of particle spectra in such a low concentration environment, the authors should acknowledge this limitation of the ATOFMS measurement in a more prominent manner (for example, Sierau 2014 acknowledges this challenge in their abstract). Related to this, what fraction of ATOFMS detected particles did not provide mass spectra? These limitations in no way contradict the main conclusions of the work, since the CCSEM-EDX analysis provides a complementary data set that provides similar conclusions; however, this limitation needs to be clearly stated.

**We acknowledge the limitations of having a small number of particle spectra, which is why much of the in-depth analysis focuses on CCSEM-EDX data. However, the main difference is that the ATOFMS used by Sierau et al. (2014) was operating properly, to our knowledge, and that paper relied solely on the ATOFMS data obtained. In contrast, during our study, there was an instrumental issue with the time-of-flight mass analyzer, as described on P8 L25- P9 L1. This resulted in an extremely low fraction of particles producing mass spectra. The mass analyzer was fixed following the field campaign, and laboratory tests verified that the mass spectra produced during the field campaign were accurate, with just a much lower fraction of individual particle mass spectra collected. In our more recent Arctic deployment of the ATOFMS (after fixing the mass analyzer), over 33,000 particles were chemically analyzed over a similar time frame as the present study (unpublished work). To more clearly acknowledge the limitations of the ATOFMS results in the present study, we now state the**

**total number of ATOFMS particles chemically analyzed in the Figure 5 caption, in addition to the statement in the methods section (P8 L23).**

P9 up to L25: Emission of small, primary particles from Prudhoe Bay emissions is likely also associated with emission of oxidizable or condensible gas phase species. Why might the Prudhoe emissions have stayed relatively small during transport to your measurement site?

**We reorganized this section (moved later paragraph discussing emissions of condensable gases up to P10 L13-20) and clarified the discussion. P10 L23-28 now reads "Kolesar et al. (2017) previously observed Prudhoe Bay air masses to preferentially exhibit particle growth, compared to Arctic Ocean air masses. However, particle growth was not observed to occur within all Prudhoe Bay air masses during the summer, and particle growth events were not observed in September in Utqiaġvik." Given the lack of measurements of trace gases both in Kolesar et al. (2017) and this study, it is not possible, unfortunately, to comment on the reason for the lack of observed particle growth. Note that Kolesar et al. (2017) did not observe any particle growth events in September in Utqiaġvik over multiple years.**

P10 L8: Is comparison to Alert, Nunavut also possible?

**We added a comparison to Alert on P11 L7. The full sentence now reads: "The median particle concentration within Arctic Ocean air masses is similar to the median particle number concentrations during August at Station Nord, Greenland (227 particles cm$^{-3}$, Nguyen et al., 2016) and Alert, Canada (~160 particles cm$^{-3}$; Croft et al., 2016), during September at Tiksi, Russia (222 particles cm$^{-3}$; Asmi et al., 2016), and within the range of observations onboard the Swedish icebreaker Oden from July – September during multiple central Arctic Ocean studies when the air masses were exposed to the open ocean (90-210 particles cm$^{-3}$; Heintzenberg et al., 2015).".**

P10 L20: However, your measurements show that the chemical composition of particles >100nm is different between the two types of air masses, and I doubt that no influence on these particle sizes occurs.

**While the aerosol size distributions above 100 nm were not statistically different between Prudhoe Bay and Arctic Ocean air masses, there were indeed chemical composition differences observed by ATOFMS and SEM-EDX (e.g. increased chloride depletion, coupled with nitrate and sulfate addition in sea spray aerosol). Since these differences in chemical composition are discussed in the later sections, this confusing sentence has been deleted.**

P11 L1: This work is from Summit, Greenland, which is arguably quite different from other, lower altitude, regions of the Arctic for a few reasons. First, ascent over Greenland can cause deposition of transported species leading to quite clean air masses. Second, the lack of local sources (aside from snow pack photochemistry) compared to other Arctic regions which are directly subject to marine, coastal, and tundra influences. While some recent studies have suggested the presence of condensible material in the summer Arctic (e.g., Mungall 2017, Willis 2016), it would be very surprising if Prudhoe Bay did not also emit gas phase species. The authors might want to discuss what is known about gas phase emissions from oil and gas extraction. Some studies from lower latitudes (e.g., Liggio 2016 ) suggest these can be substantial.

**We removed the reference to Ziemba et al. (2010) for work at Summit. These sentences (moved to P10 L17-23 in response to an earlier comment) now read: "Based on the simulations by Fierce et al. (2015), particle growth during transport for particles ~30-50 nm would take ~1-7 days, if coagulation-dominated due to limited condensable material. Particle growth was not observed during this study, suggesting that sufficient condensable material was not available for an observable change in particle diameter. Therefore, particles of this size could potentially be transported from Prudhoe Bay to Utqiaġvik during the average 21 ± 7 h transit time. Given the lack of primary ultrafine aerosol sources between Utqiaġvik and Prudhoe Bay, it is suggested that these particles were likely transported from Prudhoe Bay."**

P11 L5: A contradiction of your observations by those of Kolesar 2017 does not help to make your argument clearer here. If particle growth from Prudhoe Bay was observed previously what differences do we expect in late summer 2015? Was the time of year significant (Kolesar 2017 shows growth peaking in June to July)? What the meteorology different? This really needs further discussion.

**Our results do not contradict those of Kolesar et al. (2017), as particle growth was not observed to occur within all air masses from Prudhoe Bay in the summer. As shown in Fig. 3 of Kolesar et al. (2017), particle growth events were not observed in September, with particle growth occurring only part of the time during August. We expanded on this discussion on P10 L23-27.**

P11 L20: Given that the ATOFMS misses sulfur-rich particles, which should be a reasonable contributor to Arctic Ocean air masses owing to DMS oxidation, how reliable are ATOFMS estimates of the fractional particle number contribution from various particle types? Is there some was to estimate the fraction of particles that are detected by the ATOFMS but not desorbed and ionized due to being sulfate rich? Table 1 would suggest a large fraction of particles contained sulfate. What fraction of particles scattered light in the ATOFMS, but did not produce mass spectra?

**In order to further address this "missing" particle type, we added estimates of fractional particle number contributions based on the CCSEM-EDX observations of the sulfur-rich particles. The sentence on P12 L9-13 now states "Based on CCSEM-EDX analysis (Figure 6), these sulfur particles likely comprised ~10 – 30% of the 0.13 – 1 μm particle number fraction during Arctic Ocean air mass influence, and ~10 – 20% of the 0.13 – 0.3 μm particle number fraction during Prudhoe Bay air mass influence. Accounting for these sulfur**

**particles would reduce the reported ATOFMS fractions by ~5 – 15% for Arctic Ocean air mass influence, and ~5 – 10% for Prudhoe Bay air mass influence."**

P11 L23: Would these sources not have been captured during the 'local' or Utqiagvik influenced periods?

**As shown in Figure 5 and now stated on P12 L19-20, the contributions from biomass burning and dust particles were greatest in the wind direction of Utqiaġvik influence.**

P12 section 3.2.1: Besides trajectories and wind-sector analysis, what chemical characteristics do the authors have for the Arctic Ocean sector being representative of a clean marine background for the local region? The presence of BC-containing particles as well as aged SSA showing nitrate peaks (suggesting NOx chemistry), while I acknowledge that sulfate peaks could arise from interaction with DMS oxidation products, suggest a pollution influence (at least to some extent) on these air masses as well. Do these air masses, either at the measurement site or at the Barrow Observatory, conform the the thresholds in BC (or absorbing aerosol) usually associated with clean marine conditions (e.g., < 50 ng/m3 of BC (Gantt 2013))?

**As noted in Sec. 3.2.1 (P12 L24 – P13 L3) and shown in Fig. 6, nearly all of the supermicron particles were sea spray aerosol (SSA). As shown in Table 1, during the Arctic Ocean influence, Cl/Na ratios of the observed submicron and supermicron SSA particles were 0.81 and 0.99, respectively, indicative of fresh SSA and supporting a clean marine background; corresponding SSA S/Na ratios were also close to the expected seawater ratio (Keene et al., 1986). Previous remote marine studies have also measured some anthropogenic influences (Pósfai et al., 1999; Shank et al., 2012). Also, as discussed in Section 3.1, average particle number concentrations during the Arctic Ocean influence were comparable to other Arctic background sites, further suggesting that our measurements are representative of background Arctic conditions.**

P13 L10: Do these estimates still correspond to 2004, or rather a yearly average?

**These estimates still correspond to 2004. The sentence has been clarified: "For air masses influenced by Prudhoe Bay emissions, increased number fractions of soot, OC, and partially aged SSA particles were measured, with increased soot and OC particles expected based on 2004 estimates of soot (1.9 kt) and OC (2.0 kt) emissions from US Arctic (primarily Prudhoe Bay) oil and gas extraction activities (Peters et al., 2011)."**

P13 L17: Is this single fragment really an unambiguous indicator of SOA formation? If SOA formation was occurring on Prudhoe bay emissions, why did the particles remain quite small, or put another way why do the authors suggest earlier in the text that particle growth did not occur?

**Previous ATOFMS studies (Qin et al., 2012) established $m/z$ 43 ($C_2H_3O^+$) as a marker of oxidized organic compounds. However, since particle growth was not observed, we added the following sentence on P14 L12-13: "However, as particle growth was not observed during Prudhoe Bay air mass influence (Section 3.1), it is likely that SOA contributions to particle mass were minor."**

P13 L27: It is difficult to draw conclusions from these differences, especially when the number of detected particles is so small. Do long term aerosol absorption data from the Barrow Observatory help with these conclusions at all?

**We incorporated data from a co-located aethalometer and revised the sentence now on P14 L21-24 as follows: "Soot was also identified by ATOFMS during Prudhoe Bay periods by $C_n^+$ clusters ($C^+$ [$m/z$ 12], $C_2^+$ [$m/z$ 24], $C_3^+$ [$m/z$ 36], etc). Elevated black carbon mass concentrations (up to 0.27 μg/m$^3$) were also measured by the aethalometer during the Prudhoe Bay air mass on August 25 (Figure S5)." Unfortunately, aerosol absorption data from the Barrow Observatory are unavailable during the Prudhoe Bay influenced periods we chemically characterized in late September.**

P14 L16: Why is the main sulfate peak used to identify sulfur specifc different here compared to line 20 on the previous page?

**This is due to spectral interference between $HSO_4^-$ and $NaCl_2^-$ in SSA. We have clarified this by adding the following sentence to P12 L3-5: "Sulfate is identified as $SO_3^-$ ($m/z$ -80) in SSA due to mass spectral interferences between $HSO_4^-$ [$m/z$ -97] and $NaCl_2^-$ [$m/z$ -93,95,97] (Qin et al., 2012; Sultana et al., 2017)."**

P14 L20: Does a peak at NO2- indicate nitrite or just fragmentation of nitrate?

**Based on previous ATOFMS studies, this is likely a fragment of nitrate (e.g. Dall'Osto et al., 2009; Liu et al., 2003). The nitrite identification has been removed.**

Figure 1: Average trajectories for the study period might be more meaningful that selected trajectories. Additionally, is the area North of Utqiagvik completely ice free during the measurement, as pictured?

**As suggested, we replaced the representative trajectories with average trajectories. The area north of Utqiagvik, shown in the Figure, was completely ice free during the study, as pictured.**

Figure 2: Percentiles to illustrate the range of the data might be more appropriate here

**We added Figure S3, which shows the 25th, 50th and 75th percentiles for the Prudhoe Bay and Arctic Ocean particle number distributions. A note was also added to the Figure 2 caption referring to this new supplemental figure.**

Figure 4: Were elemental carbon peaks observed in OC particles? And similarly, was OC present on the EC particles detected? Can carbon and oxygen peaks be quantified in the CCSEM-EDX spectra?

**OC markers were not observed in the EC particle mass spectra. However, minor contributions from m/z 12 ($C^+$) were observed in the OC particle mass spectra (now noted in the Supplemental P1 L20); however, without evidence of many carbon cluster ions, this ion peak can also be attributed as an OC fragment ion. However, the EC peaks in the OC mass spectra were minor and therefore not labeled in Figure 4 due to space constraints.**

**Unfortunately, there are substrate interferences with carbon and oxygen peaks in the CCSEM-EDX spectra.**

Figure 5: The number of detected particles should be noted in this figure caption

**This is now noted.**

Supplement: Is mention of the TMA containing particles warranted in the main body of the paper? It helps to show different source influences, since these were presumably detected during Arctic Ocean influence. Or, are there too few particle spectra with TMA peaks to draw conclusions?

**Unfortunately, there were too few TMA-containing mass spectra measured to draw conclusions. We added a sentence to Supplemental P1 L26 indicating this.**

Specific comments:

P3 L4: natural gas

**We added this.**

P3 L7: Which types of pollutants are relevant here? Maybe list the major ones we expect, that are then discussed in following paragraphs?

**We now list PM, VOCs, SO$_2$, and NO$_x$ here.**

P4 L13: "primary aerosol can. . ."

**We added this.**

P4 L22: What is meant by "mostly" here?

**We replaced "mostly" with "primarily".**

P9 L17: perhaps: "Arctic Ocean influenced periods"

**We changed this to "Arctic Ocean influenced air masses".**

P9 L19: Prudhoe Bay air masses

**We've changed this wording to Prudhoe Bay air masses.**

P15 L26: respectively

**We've corrected this word.**

P16 L5: "may contribute to further decrease"

**This has been added.**

**References**

Arnold, S., Law, K., Thomas, J., Starckweather, S., von Salzen, K., Stohl, A., Sharma, S., Lund, M., Flanner, M., Petäjä, T., 2016. Arctic air pollution. Elementa: Science of the Anthropocene.

Asmi, E., Kondratyev, V., Brus, D., Laurila, T., Lihavainen, H., Backman, J., Vakkari, V., Aurela, M., Hatakka, J., Viisanen, Y., 2016. Aerosol size distribution seasonal characteristics measured in Tiksi, Russian Arctic. Atmos. Chem. Phys. 16, 1271-1287.

Barrett, T., Robinson, E., Usenko, S., Sheesley, R., 2015. Source contributions to wintertime elemental and organic carbon in the western arctic based on radiocarbon and tracer apportionment. Environ. Sci. Technol. 49, 11631-11639.

Bond, T.C., Doherty, S.J., Fahey, D., Forster, P., Berntsen, T., DeAngelo, B., Flanner, M., Ghan, S., Kärcher, B., Koch, D., 2013. Bounding the role of black carbon in the climate system: A scientific assessment. J. Geophys. Res-Atmos. 118, 5380-5552.

Brock, C.A., Cozic, J., Bahreini, R., Froyd, K.D., Middlebrook, A.M., McComiskey, A., Brioude, J., Cooper, O., Stohl, A., Aikin, K., 2011. Characteristics, sources, and transport of aerosols measured in spring 2008 during the aerosol, radiation, and cloud processes affecting Arctic Climate (ARCPAC) Project. Atmos. Chem. Phys. 11, 2423-2453.

Cappa, C.D., Onasch, T.B., Massoli, P., Worsnop, D.R., Bates, T.S., Cross, E.S., Davidovits, P., Hakala, J., Hayden, K.L., Jobson, B.T., 2012. Radiative absorption enhancements due to the mixing state of atmospheric black carbon. Science 337, 1078-1081.

Chung, S.H., Seinfeld, J.H., 2005. Climate response of direct radiative forcing of anthropogenic black carbon. J. Geophys. Res-Atmos. 110.

Croft, B., Martin, R.V., Leaitch, W.R., Tunved, P., Breider, T.J., D'Andrea, S.D., Pierce, J.R., 2016. Processes controlling the annual cycle of Arctic aerosol number and size distributions. Atmos. Chem. Phys. 16, 3665-3682.

Dall'Osto, M., Harrison, R., Coe, H., Williams, P., Allan, J., 2009. Real time chemical characterization of local and regional nitrate aerosols. Atmos. Chem. Phys. 9, 3709-3720.

Fierce, L., Riemer, N., Bond, T.C., 2015. Explaining variance in black carbon's aging timescale. Atmos. Chem. Phys. 15, 3173-3191.

Flanner, M.G., 2013. Arctic climate sensitivity to local black carbon. J. Geophys. Res-Atmos. 118, 1840-1851.

Flanner, M.G., Zender, C.S., Randerson, J.T., Rasch, P.J., 2007. Present-day climate forcing and response from black carbon in snow. J. Geophys. Res-Atmos. 112.

Healy, R.M., Wang, J.M., Jeong, C.H., Lee, A.K., Willis, M.D., Jaroudi, E., Zimmerman, N., Hilker, N., Murphy, M., Eckhardt, S., 2015. Light-absorbing properties of ambient black carbon and brown carbon from fossil fuel and biomass burning sources. J. Geophys. Res-Atmos. 120, 6619-6633.

Heintzenberg, J., Leck, C., Tunved, P., 2015. Potential source regions and processes of aerosol in the summer Arctic. Atmos. Chem. Phys 15, 6487-6502.

Jacobson, M.Z., 2001. Strong radiative heating due to the mixing state of black carbon in atmospheric aerosols. Nature 409, 695-697.

Keene, W.C., Pszenny, A.A., Galloway, J.N., Hawley, M.E., 1986. Sea-salt corrections and interpretation of constituent ratios in marine precipitation. J. Geophys. Res-Atmos. 91, 6647-6658.

Knox, A., Evans, G., Brook, J., Yao, X., Jeong, C.-H., Godri, K., Sabaliauskas, K., Slowik, J., 2009. Mass absorption cross-section of ambient black carbon aerosol in relation to chemical age. Aerosol. Sci. Technol. 43, 522-532.

Koch, D., Schulz, M., Kinne, S., McNaughton, C., Spackman, J., Balkanski, Y., Bauer, S., Berntsen, T., Bond, T.C., Boucher, O., 2009. Evaluation of black carbon estimations in global aerosol models. Atmos. Chem. Phys. 9, 9001-9026.

Kolesar, K.R., Cellini, J., Peterson, P.K., Jefferson, A., Tuch, T., Birmili, W., Wiedensohler, A., Pratt, K.A., 2017. Effect of Prudhoe Bay emissions on atmospheric aerosol growth events observed in Utqiaġvik (Barrow), Alaska. Atmos. Environ. 152, 146-155.

Liu, D.Y., Wenzel, R.J., Prather, K.A., 2003. Aerosol time-of-flight mass spectrometry during the Atlanta Supersite Experiment: 1. Measurements. J. Geophys. Res-Atmos. 108.

Liu, S., Aiken, A.C., Gorkowski, K., Dubey, M.K., Cappa, C.D., Williams, L.R., Herndon, S.C., Massoli, P., Fortner, E.C., Chhabra, P.S., 2015. Enhanced light absorption by mixed source black and brown carbon particles in UK winter. Nat. Commun. 6.

Moffet, R.C., Prather, K.A., 2009. In-situ measurements of the mixing state and optical properties of soot with implications for radiative forcing estimates. Proc. Natl. Acad. Sci. 106, 11872-11877.

Nguyen, Q.T., Glasius, M., Sørensen, L.L., Jensen, B., Skov, H., Birmili, W., Wiedensohler, A., Kristensson, A., Nøjgaard, J.K., Massling, A., 2016. Seasonal variation of atmospheric particle number concentrations, new particle formation and atmospheric oxidation capacity at the high Arctic site Villum Research Station, Station Nord. Atmos. Chem. Phys. 16, 11319-11336.

Peters, G., Nilssen, T., Lindholt, L., Eide, M., Glomsrød, S., Eide, L., Fuglestvedt, J., 2011. Future emissions from shipping and petroleum activities in the Arctic. Atmos. Chem. Phys. 11, 5305-5320.

Pósfai, M., Anderson, J.R., Buseck, P.R., Sievering, H., 1999. Soot and sulfate aerosol particles in the remote marine troposphere. J. Geophys. Res-Atmos. 104, 21685-21693.

Qin, X., Pratt, K.A., Shields, L.G., Toner, S.M., Prather, K.A., 2012. Seasonal comparisons of single-particle chemical mixing state in Riverside, CA. Atmos. Environ. 59, 587-596.

Quinn, P., Miller, T., Bates, T., Ogren, J., Andrews, E., Shaw, G., 2002. A 3-year record of simultaneously measured aerosol chemical and optical properties at Barrow, Alaska. J. Geophys. Res-Atmos. 107.

Quinn, P., Shaw, G., Andrews, E., Dutton, E., Ruoho-Airola, T., Gong, S., 2007. Arctic haze: current trends and knowledge gaps. Tellus B 59, 99-114.

Sand, M., Berntsen, T.K., Kay, J.E., Lamarque, J.F., Seland, Ø., Kirkevåg, A., 2013. The Arctic response to remote and local forcing of black carbon. Atmos. Chem. Phys. 13, 211-224.

Shank, L., Howell, S., Clarke, A., Freitag, S., Brekhovskikh, V., Kapustin, V., McNaughton, C., Campos, T., Wood, R., 2012. Organic matter and non-refractory aerosol over the remote Southeast Pacific: oceanic and combustion sources. Atmos. Chem. Phys. 12, 557-576.

Sharma, S., Ishizawa, M., Chan, D., Lavoué, D., Andrews, E., Eleftheriadis, K., Maksyutov, S., 2013. 16-year simulation of Arctic black carbon: Transport, source contribution, and sensitivity analysis on deposition. J. Geophys. Res-Atmos. 118, 943-964.

Sierau, B., Chang, R.-W., Leck, C., Paatero, J., Lohmann, U., 2014. Single-particle characterization of the high-Arctic summertime aerosol. Atmos. Chem. Phys. 14, 7409-7430.

Sultana, C.M., Collins, D.B., Prather, K.A., 2017. Effect of Structural Heterogeneity in Chemical Composition on Online Single-Particle Mass Spectrometry Analysis of Sea Spray Aerosol Particles. Environ. Sci. Technol 51, 3660-3668.

Willis, R., Blanchard, F., Conner, T., 2002. Guidelines for the application of SEM/EDX analytical techniques to particulate matter samples. EPA. Washington, US.

Ziemba, L.D., Dibb, J.E., Griffin, R.J., Huey, L.G., Beckman, P., 2010. Observations of particle growth at a remote, Arctic site. Atmos. Environ. 44, 1649-1657.

---

## Author Comment (AC2) · 29 Jul 2017

**Reviewer #2**

General comments: In this study, the authors present observations of particle composition and size distributions from the Barrow Environmental Observatory. The authors compare and contrast observations of aerosol from air masses that originated over the Arctic Ocean with aerosol from air masses that originated from the direction of the Prudhoe Bay oil fields. I recommend the manuscript for publication, provided that my following points, and those of the first reviewer, are addressed. In order to reduce redundancy for the authors' response, I will restrict my points to those not covered by the first reviewer.

Specific comments:

P3 Lines 14-16: Stohl et al. (2013) makes no attribution of PM or OC from Prudhoe Bay.

> **We clarified this statement to now read: "The majority of PM emitted by US Arctic oil and gas extraction sources (turbine gas combustion, diesel emissions from generators and vehicles, and flaring (Stohl et al., 2013)) in 2004 corresponded to BC (1.9 kt) and OC (2.0 kt) (Peters et al., 2011)".**

P9 Lines 18-25: I think perhaps that the authors intend to exclude regional-scale nucleation events here, but not nucleation within emissions plumes from the Prudhoe Bay oil fields. The authors even suggest that the emissions from such drilling operations can contribute to new-particle formation on page 3, lines 19-21. Observations of particle growth would be expected for a regional-scale nucleation event, but not for continuous nucleation within an emitted plume of condensable gases, such as those observed by Brock et al. (2002) and Brock et al. (2003). If there was continuous new-particle formation occurring within a plume from the Prudhoe Bay oil field, under steady-state conditions (including constant wind speed and direction), no change in the size of observed aerosol would be observed unless the distance from the source changed. The age of the aerosol between nucleation in the plume and detection at the observation site would be constant with time, and therefore so would the size of the aerosol.

> **We clarified this on P10 L9-10: "However, regional new particle formation would typically be followed by particle growth (Kulmala et al., 2004), which was not observed (Figure S2)."**

Technical Corrections:
P11 Line 16: "(Sierau et al. (2014)" should be "(Sierau et al., 2014)"

> **We corrected this.**

**References**

Kulmala, M., Vehkamäki, H., Petäjä, T., Dal Maso, M., Lauri, A., Kerminen, V.-M., Birmili, W., and McMurry, P. H.: Formation and growth rates of ultrafine atmospheric particles: a review of observations, J. Aerosol. Sci., 35, 143-176, 2004.

Peters, G., Nilssen, T., Lindholt, L., Eide, M., Glomsrød, S., Eide, L., and Fuglestvedt, J.: Future emissions from shipping and petroleum activities in the Arctic, Atmos. Chem. Phys., 11, 5305-5320, 2011.

Stohl, A., Klimont, Z., Eckhardt, S., Kupiainen, K., Shevchenko, V., Kopeikin, V., and Novigatsky, A.: Black carbon in the Arctic: the underestimated role of gas flaring and residential combustion emissions, Atmos. Chem. Phys., 13, 8833-8855, 2013.

---

## Author Response (AR1)

**Response to Reviewer #1**

**We thank the reviewer for their helpful comments and suggestions. We provide below the original comments, shown in gray, and our responses, with specific revisions noted, in bold font.**

This manuscript describes single particle characterization (ATOFMS and CCSEM- EDX) of particulate matter attributed to Arctic oil extraction activities at Prudhoe Bay, Alaska, and contrasts this with particulate matter dominated by natural emissions from the nearby Arctic Ocean for ~1 month of measurements in late summer. This work con- tributes to the very few studies of local Arctic emissions of anthropogenic aerosol, and is therefore valuable in extending our understanding of local Arctic pollution sources in the context of the summertime Arctic natural background. The manuscript is overall well written and clear. It merits publication in ACP after the following comments have been addressed.

**Major Comments:**

In the introduction as well as in the discussion sections, the authors should make further efforts to ensure that proper, complete, and most importantly accurate, credit is given to prior related work. Specific instances are indicated in the minor comments.

**We added and revised references used throughout, with specific instances requested by the reviewer addressed in the responses to the comments below.**

**Minor Comments:**

Introduction: This introduction gives no context for the conditions expected in the unpolluted summer Arctic. The seasonal cycle in Arctic aerosol is very relevant to the context of these measurements, especially under the cleaner conditions of summer to autumn. Similarly, an acknowledgment of pollution influences in winter and spring is also relevant. The natural question here is how Prudhoe Bay emissions might compare to influences of long range transport in terms of aerosol loading and composition.

We added a statement to the introduction (P3 L9-13) addressing pollution contributions from oil and gas extraction in regards to the seasonal cycle, which states: "The Arctic aerosol population is characterized by a maximum mass loading in the winter, due to transported pollutants from the mid-latitudes, and a minimum in the summer, when local sources, including sea spray aerosol, dominate (Quinn et al., 2002; Quinn et al., 2007). However, there is limited knowledge of aerosols produced within the Arctic, particularly in the context of changing emissions from both natural and anthropogenic sources (Arnold et al., 2016)."

P3 L10-15: The discussion of BC and OC contributions from total Arctic oil and gas extraction, versus that from Prudhoe Bay is at present somewhat confusing. In addition, the methods used to arrive at BC and OC estimates might be relevant to this discussion, e.g., were in-situ measurements used to arrive at these estimates?

We revised this paragraph to reduce confusion about what is from the overall Arctic vs. Prudhoe Bay.

P3 L17: Are these US contributions from Prudhoe Bay alone or does this estimate represent expansion to other sites?

**These particulate emissions represent contributions from all US Arctic oil and gas extraction sites. We clarified in this paragraph that Prudhoe Bay is the primary, but not only, contributor to the estimated US oil field emissions in the Arctic.**

P3 L20: It is not strictly true to suggest that emission of gas phase species would lead to solely new particle formation. If the intention is to highlight the role that the very low condensation sinks of the summer Arctic could play in this respect, then the authors should state this clearly.

**We revised this to state "...drilling operations can emit aerosol precursors (NOx, SO2, and VOCs) and alter oxidant levels, which can lead to the formation of secondary aerosol, as well as contribute to new particle formation...".**

P3 L22: Flanner et al., 2013 does not indicate anything about the impact of BC on clouds. In abstract the authors state that the study has limitations owing to their lack of treatment of indirect effects.

**This was not clearly worded, and we clarified this to now state: "BC is estimated to have a warming effect on the Arctic atmosphere (e.g. Bond et al., 2013; Flanner, 2013; Flanner et al., 2007; Sand et al., 2013; Sharma et al., 2013)."**

P3 L25: Citation of other relevant work, such as Flanner et al, 2007 (and others) is advisable here

**References to Flanner et al. (2007), Sand et al. (2013), and Sharma et al. (2013) have been added to this paragraph.**

P3 L27: Is this result directly from Koch 2009, or is it elaborated by Bond 2013?

**This is directly from Koch et al. (2009), and we updated the sentence to accurately reflect this.**

P3 L27-28: Did the modelling studies discussed here include local Arctic BC sources? This is worth discussing.

The following sentence was added on P4 L4-7 to discuss that these modeling inaccuracies may be improved by the inclusion of local Arctic BC sources. "Koch et al. (2009) suggest that Arctic BC concentrations are under-predicted by a variety of models by an average factor of 2.5, which may be improved by more accurately incorporating local BC sources (Flanner, 2013)."

P4 L4: To do justice to the study of Barrett 2015 the authors should discuss the varying contribution of fossil fuels to Barrow EC over the winter period. Biomass burning is also an important source and can have local Arctic sources.

**Following the reviewer's suggestion, we expanded the discussion of Barrett et al. (2015) (P4 L9-11) to now state: "...similar to the recent results of Barrett et al. (2015) at Utqiaġvik during December 2012 – March 2013 when transported particles from Russian oil fields were observed. Barrett et al. (2015) also measured regional Arctic BC from both fossil fuel combustion and biomass burning."**

P4 L9: Do the authors suggest that due to the focus on biomass burning, the results of this study are less relevant to their discussion? The measurements of Brock 2011 likely represent some of the few more complete chemical characterizations of Prudhoe Bay aerosol emissions.

**We removed this statement; it was not our intention to appear to be overlooking the characterization efforts of Brock et al. (2011) in terms of the Prudhoe Bay aerosol.**

P4 L22: Cappa 2012 explicitly shows that absorption enhancement was not detectable, despite significant particle mixing, during their measurement campaign. This needs to be much more clearly stated, and the complexity of this issue need to be addressed. In particular, I do not agree with the statement that light absorption is enhanced the presence of sulfate or organic species in the same particles. Some studies have observed absorption enhancements (e.g., Knox 2009, Liu 2015) while others have not (e.g., Cappa 2012, Healy 2015). It is not clear whether this effect is significant in all ambient aerosol and this issue should not be stated as being entirely clear.

**We acknowledge that this is a complex topic and updated P4 L26-28 accordingly: "It is currently not clear whether light absorption by a BC particle is enhanced by sulfate or organic coatings (e.g., Cappa et al., 2012; Chung and Seinfeld, 2005; Healy et al., 2015; Jacobson, 2001; Knox et al., 2009; Liu et al., 2015; Moffet and Prather, 2009)."**

P4 L26: Is such a binary definition relevant to ambient particles? It might be more accurate to state that ambient aerosol can be found with range of mixing states approaching external or internal mixing in different cases and with influence from different source types.

**We clarified this on P5 L3-5, which now reads: "Ambient aerosol populations typically vary between internal mixtures, with multiple chemical species contained within a single particle, and external mixtures, with chemical species present as separate particles (Prather et al., 2008)."**

Method section: Please ensure that all manufacturer information is as complete as possible with relevant names, models and locations.

**This information has been added.**

**P5 L21: How do back trajectories compare for the two sites?**

**These sites are separated by 5.5 km, less than the resolution of HYSPLIT (1 degree, ~100 km), with only flat tundra located between them (P6 L1). Therefore, use of HYSPLIT to differentiate between the sites is not possible, and trajectories at the two sites are expected to be similar.**

P6 L23: Given the long sampling period and the acknowledgement of local vehicle emission contributing to particle measurements, how might fast-varying local emissions (such as vehicles or generators) influence the particle composition interpreted from these 8h measurements?

**Short particle spikes due to local emissions would have been detected as brief increases in particle number concentrations by the SMPS, operated with 5 min resolution (Figure S2). These events were only observed during the Utqiagvik influence sampling periods (e.g. Sept 7, see Figure S2); these local contributions were not observed during the clean Arctic or Prudhoe Bay influenced periods (Figure S2).**

Section 2.3: How is statistical significance determined for these measurements? What is the minimum number of particles that must be analyzed in order to to have a representative sample?

**Significance was determined using binomial statistics, which we have clarified on P8 L5-6. The minimum number of particles for a representative sample are between ~300 and 1,000. See Willis et al. (2002), Section 4.6.2.**

P8 L10: This is a remarkably small number of particle spectra to draw conclusions from. While I acknowledge the challenges of obtaining a large number of particle spectra in such a low concentration environment, the authors should acknowledge this limitation of the ATOFMS measurement in a more prominent manner (for example, Sierau 2014 acknowledges this challenge in their abstract). Related to this, what fraction of ATOFMS detected particles did not provide mass spectra? These limitations in no way contradict the main conclusions of the work, since the CCSEM-EDX analysis provides a complementary data set that provides similar conclusions; however, this limitation needs to be clearly stated.

We acknowledge the limitations of having a small number of particle spectra, which is why much of the in-depth analysis focuses on CCSEM-EDX data. However, the main difference is that the ATOFMS used by Sierau et al. (2014) was operating properly, to our knowledge, and that paper relied solely on the ATOFMS data obtained. In contrast, during our study, there was an instrumental issue with the time-of-flight mass analyzer, as described on P8 L25- P9 L1. This resulted in an extremely low fraction of particles producing mass spectra. The mass analyzer was fixed following the field campaign, and laboratory tests verified that the mass spectra produced during the field campaign were accurate, with just a much lower fraction of individual particle mass spectra collected. In our more recent Arctic deployment of the ATOFMS (after fixing the mass analyzer), over 33,000 particles were chemically analyzed over a similar time frame as the present study (unpublished work). To more clearly acknowledge the limitations of the ATOFMS results in the present study, we now state the

**total number of ATOFMS particles chemically analyzed in the Figure 5 caption, in addition to the statement in the methods section (P8 L23).**

P9 up to L25: Emission of small, primary particles from Prudhoe Bay emissions is likely also associated with emission of oxidizable or condensible gas phase species. Why might the Prudhoe emissions have stayed relatively small during transport to your measurement site?

We reorganized this section (moved later paragraph discussing emissions of condensable gases up to P10 L13-20) and clarified the discussion. P10 L23-28 now reads "Kolesar et al. (2017) previously observed Prudhoe Bay air masses to preferentially exhibit particle growth, compared to Arctic Ocean air masses. However, particle growth was not observed to occur within all Prudhoe Bay air masses during the summer, and particle growth events were not observed in September in Utqiaġvik." Given the lack of measurements of trace gases both in Kolesar et al. (2017) and this study, it is not possible, unfortunately, to comment on the reason for the lack of observed particle growth. Note that Kolesar et al. (2017) did not observe any particle growth events in September in Utqiaġvik over multiple years.

**P10 L8: Is comparison to Alert, Nunavut also possible?**

We added a comparison to Alert on P11 L7. The full sentence now reads: "The median particle concentration within Arctic Ocean air masses is similar to the median particle number concentrations during August at Station Nord, Greenland (227 particles cm-3, Nguyen et al., 2016) and Alert, Canada (~160 particles cm-3; Croft et al., 2016), during September at Tiksi, Russia (222 particles cm-3; Asmi et al., 2016), and within the range of observations onboard the Swedish icebreaker Oden from July – September during multiple central Arctic Ocean studies when the air masses were exposed to the open ocean (90-210 particles cm-3; Heintzenberg et al., 2015).".

P10 L20: However, your measurements show that the chemical composition of particles >100nm is different between the two types of air masses, and I doubt that no influence on these particle sizes occurs.

While the aerosol size distributions above 100 nm were not statistically different between Prudhoe Bay and Arctic Ocean air masses, there were indeed chemical composition differences observed by ATOFMS and SEM-EDX (e.g. increased chloride depletion, coupled with nitrate and sulfate addition in sea spray aerosol). Since these differences in chemical composition are discussed in the later sections, this confusing sentence has been deleted. P11 L1: This work is from Summit, Greenland, which is arguably quite different from other, lower altitude, regions of the Arctic for a few reasons. First, ascent over Greenland can cause deposition of transported species leading to quite clean air masses. Second, the lack of local sources (aside from snow pack photochemistry) compared to other Arctic regions which are directly subject to marine, coastal, and tundra influences. While some recent studies have suggested the presence of condensible material in the summer Arctic (e.g., Mungall 2017, Willis 2016), it would be very surprising if Prudhoe Bay did not also emit gas phase species. The authors might want to discuss what is known about gas phase emissions from oil and gas extraction. Some studies from lower latitudes (e.g., Liggio 2016 ) suggest these can be substantial.

We removed the reference to Ziemba et al. (2010) for work at Summit. These sentences (moved to P10 L17-23 in response to an earlier comment) now read: "Based on the simulations by Fierce et al. (2015), particle growth during transport for particles ~30-50 nm would take ~1-7 days, if coagulation-dominated due to limited condensable material. Particle growth was not observed during this study, suggesting that sufficient condensable material was not available for an observable change in particle diameter. Therefore, particles of this size could potentially be transported from Prudhoe Bay to Utqiaġvik during the average 21  $\pm$  7 h transit time. Given the lack of primary ultrafine aerosol sources between Utqiaġvik and Prudhoe Bay, it is suggested that these particles were likely transported from Prudhoe Bay."

P11 L5: A contradiction of your observations by those of Kolesar 2017 does not help to make your argument clearer here. If particle growth from Prudhoe Bay was observed previously what differences do we expect in late summer 2015? Was the time of year significant (Kolesar 2017 shows growth peaking in June to July)? What the meteorology different? This really needs further discussion.

Our results do not contradict those of Kolesar et al. (2017), as particle growth was not observed to occur within all air masses from Prudhoe Bay in the summer. As shown in Fig. 3 of Kolesar et al. (2017), particle growth events were not observed in September, with particle growth occurring only part of the time during August. We expanded on this discussion on P10 L23-27.

P11 L20: Given that the ATOFMS misses sulfur-rich particles, which should be a reasonable contributor to Arctic Ocean air masses owing to DMS oxidation, how reliable are ATOFMS estimates of the fractional particle number contribution from various particle types? Is there some was to estimate the fraction of particles that are detected by the ATOFMS but not desorbed and ionized due to being sulfate rich? Table 1 would suggest a large fraction of particles contained sulfate. What fraction of particles scattered light in the ATOFMS, but did not produce mass spectra?

In order to further address this "missing" particle type, we added estimates of fractional particle number contributions based on the CCSEM-EDX observations of the sulfur-rich particles. The sentence on P12 L9-13 now states "Based on CCSEM-EDX analysis (Figure 6), these sulfur particles likely comprised ~10 – 30% of the 0.13 – 1  $\mu$ m particle number fraction during Arctic Ocean air mass influence, and ~10 – 20% of the 0.13 – 0.3  $\mu$ m particle number fraction during Prudhoe Bay air mass influence. Accounting for these sulfur

**particles would reduce the reported ATOFMS fractions by $\sim 5 - 15\%$ for Arctic Ocean air mass influence, and $\sim 5 - 10\%$ for Prudhoe Bay air mass influence."**

P11 L23: Would these sources not have been captured during the 'local' or Utqiagvik influenced periods?

**As shown in Figure 5 and now stated on P12 L19-20, the contributions from biomass burning and dust particles were greatest in the wind direction of Utqiaġvik influence.**

P12 section 3.2.1: Besides trajectories and wind-sector analysis, what chemical characteristics do the authors have for the Arctic Ocean sector being representative of a clean marine background for the local region? The presence of BC-containing particles as well as aged SSA showing nitrate peaks (suggesting NOx chemistry), while I acknowledge that sulfate peaks could arise from interaction with DMS oxidation products, suggest a pollution influence (at least to some extent) on these air masses as well. Do these air masses, either at the measurement site or at the Barrow Observatory, conform the the thresholds in BC (or absorbing aerosol) usually associated with clean marine conditions (e.g., < 50 ng/m3 of BC (Gantt 2013))?

As noted in Sec. 3.2.1 (P12 L24 – P13 L3) and shown in Fig. 6, nearly all of the supermicron particles were sea spray aerosol (SSA). As shown in Table 1, during the Arctic Ocean influence, Cl/Na ratios of the observed submicron and supermicron SSA particles were 0.81 and 0.99, respectively, indicative of fresh SSA and supporting a clean marine background; corresponding SSA S/Na ratios were also close to the expected seawater ratio (Keene et al., 1986). Previous remote marine studies have also measured some anthropogenic influences (Pósfai et al., 1999; Shank et al., 2012). Also, as discussed in Section 3.1, average particle number concentrations during the Arctic Ocean influence were comparable to other Arctic background sites, further suggesting that our measurements are representative of background Arctic conditions.

P13 L10: Do these estimates still correspond to 2004, or rather a yearly average?

These estimates still correspond to 2004. The sentence has been clarified: "For air masses influenced by Prudhoe Bay emissions, increased number fractions of soot, OC, and partially aged SSA particles were measured, with increased soot and OC particles expected based on 2004 estimates of soot (1.9 kt) and OC (2.0 kt) emissions from US Arctic (primarily Prudhoe Bay) oil and gas extraction activities (Peters et al., 2011)."

P13 L17: Is this single fragment really an unambiguous indicator of SOA formation? If SOA formation was occurring on Prudhoe bay emissions, why did the particles remain quite small, or put another way why do the authors suggest earlier in the text that particle growth did not occur?

Previous ATOFMS studies (Qin et al., 2012) established m/z 43 (C2H3O+) as a marker of oxidized organic compounds. However, since particle growth was not observed, we added the following sentence on P14 L12-13: "However, as particle growth was not observed during Prudhoe Bay air mass influence (Section 3.1), it is likely that SOA contributions to particle mass were minor."

P13 L27: It is difficult to draw conclusions from these differences, especially when the number of detected particles is so small. Do long term aerosol absorption data from the Barrow Observatory help with these conclusions at all?

We incorporated data from a co-located aethalometer and revised the sentence now on P14 L21-24 as follows: "Soot was also identified by ATOFMS during Prudhoe Bay periods by  $C_n^+$  clusters (C+ [*m*/*z* 12],  $C_2^+$  [*m*/*z* 24],  $C_3^+$  [*m*/*z* 36], etc). Elevated black carbon mass concentrations (up to 0.27 µg/m3) were also measured by the aethalometer during the Prudhoe Bay air mass on August 25 (Figure S5)." Unfortunately, aerosol absorption data from the Barrow Observatory are unavailable during the Prudhoe Bay influenced periods we chemically characterized in late September.

P14 L16: Why is the main sulfate peak used to identify sulfur specifc different here compared to line 20 on the previous page?

This is due to spectral interference between HSO4- and NaCl2- in SSA. We have clarified this by adding the following sentence to P12 L3-5: "Sulfate is identified as SO3- (m/z -80) in SSA due to mass spectral interferences between HSO4- [m/z -97] and NaCl2- [m/z -93,95,97] (Qin et al., 2012; Sultana et al., 2017)."

P14 L20: Does a peak at NO2- indicate nitrite or just fragmentation of nitrate?

Based on previous ATOFMS studies, this is likely a fragment of nitrate (e.g. Dall'Osto et al., 2009; Liu et al., 2003). The nitrite identification has been removed.

Figure 1: Average trajectories for the study period might be more meaningful that selected trajectories. Additionally, is the area North of Utqiagvik completely ice free during the measurement, as pictured?

As suggested, we replaced the representative trajectories with average trajectories. The area north of Utqiagvik, shown in the Figure, was completely ice free during the study, as pictured.

Figure 2: Percentiles to illustrate the range of the data might be more appropriate here

**We added Figure S3, which shows the 25th, 50th and 75th percentiles for the Prudhoe Bay and Arctic Ocean particle number distributions. A note was also added to the Figure 2 caption referring to this new supplemental figure.**

Figure 4: Were elemental carbon peaks observed in OC particles? And similarly, was OC present on the EC particles detected? Can carbon and oxygen peaks be quantified in the CCSEM-EDX spectra?

OC markers were not observed in the EC particle mass spectra. However, minor contributions from  $m/z \ 12 \ (C^+)$  were observed in the OC particle mass spectra (now noted in the Supplemental P1 L20); however, without evidence of many carbon cluster ions, this ion peak can also be attributed as an OC fragment ion. However, the EC peaks in the OC mass spectra were minor and therefore not labeled in Figure 4 due to space constraints.

**Unfortunately, there are substrate interferences with carbon and oxygen peaks in the CCSEM-EDX spectra.**

Figure 5: The number of detected particles should be noted in this figure caption

**This is now noted.**

Supplement: Is mention of the TMA containing particles warranted in the main body of the paper? It helps to show different source influences, since these were presumably detected during Arctic Ocean influence. Or, are there too few particle spectra with TMA peaks to draw conclusions?

**Unfortunately, there were too few TMA-containing mass spectra measured to draw conclusions. We added a sentence to Supplemental P1 L26 indicating this.**

Specific comments:

P3 L4: natural gas

**We added this.**

P3 L7: Which types of pollutants are relevant here? Maybe list the major ones we expect, that are then discussed in following paragraphs?

**We now list PM, VOCs, SO2, and NOx here.**

P4 L13: "primary aerosol can. . ."

**We added this.**

P4 L22: What is meant by "mostly" here?

**We replaced "mostly" with "primarily".**

P9 L17: perhaps: "Arctic Ocean influenced periods"

**We changed this to "Arctic Ocean influenced air masses".**

P9 L19: Prudhoe Bay air masses

**We've changed this wording to Prudhoe Bay air masses.**

P15 L26: respectively

**We've corrected this word.**

P16 L5: "may contribute to further decrease"

**This has been added.**

**Contributions of Transported Prudhoe Bay OilfieldOil Field Emissions to the Aerosol Population in Utqiaġvik, Alaska**

Matthew J. Gunsch1, Rachel M. Kirpes1, Katheryn R. Kolesar1, Tate E. Barrett2, Swarup China3, Rebecca J. Sheesley2,4, Alexander Laskin3, Alfred Wiedensohler5, Thomas Tuch5 5 Kerri A. Pratt1,6

1Department of Chemistry, University of Michigan, Ann Arbor, MI, USA 2The Institute of Ecological, Earth, and Environmental Sciences, Baylor University, Waco, TX, USA 3Environmental Molecular Sciences Laboratory, Pacific Northwest National Laboratory, Richland, WA, USA

4Department of Environmental Science, Baylor University, Waco, TX, USA
 5Leibniz Institute for Tropospheric Research, Leipzig, Germany
 6Department of Earth and Environmental Sciences, University of Michigan, Ann Arbor, MI, USA

Correspondence to: Kerri A. Pratt (prattka@umich.edu)

Abstract. Loss of sea ice is opening the Arctic to increasing development involving oil and gas extraction

- 15 and shipping. Given the significant impacts of absorbing aerosol and secondary aerosol precursors emitted within the rapidly warming Arctic region, there is a need to characterize local anthropogenic aerosol sources and compare to natural conditions. From August-September 2015 in Utqiaġvik, AK, the chemical composition of individual atmospheric particles was measured by computer-controlled scanning electron microscopy with energy dispersive X-ray spectroscopy (0.13 - 4 µm projected area diameter)
- 20 and real-time single particle mass spectrometry  $(0.2 1.5 \ \mu\text{m} \text{ aerodynamic diameter})$ . During Arctic Ocean influenced periods (70% of the study), our results show that fresh sea spray aerosol contributed ~20%, by number, of particles between  $0.13 0.4 \ \mu\text{m}$ , 40 70% between  $0.4 1 \ \mu\text{m}$ , and 80 100% of  $1 4 \ \mu\text{m}$  particles. In contrast, for periods influenced by emissions from Prudhoe Bay (10% of the study), the third largest oilfieldoil field in North America, there was a strong influence from submicron (0.13 -
- $1 \mu m$ ) combustion derived particles (20 50% OC, by number, 5 10% soot by number). While sea spray aerosol still comprised a large fraction of particles (90% by number from  $1 - 4 \mu m$ ) detected under Prudhoe Bay influence, these particles were internally mixed with sulfate and nitrate indicative of aging processes during transport. In addition, the overall mode of the particle size number distribution shifted

from 76 nm during Arctic Ocean influence to 27 nm during Prudhoe Bay influence with particle concentrations increasing from 130 cm-3 to 920 cm-3 due to transported particle emissions from the oil fields. The increased contributions of carbonaceous combustion products and partially aged SSAsea spray aerosol should be taken into consideration forconsidered in future Arctic atmospheric composition and 5 climate simulations.

**1** Introduction**

The Arctic is experiencing dramatic climate change with sea ice extent declining rapidly and complete summertime sea ice loss expected by 2050 (Wang and Overland, 2015; Overland and Wang, 2013). With 30% of the world's undiscovered natural-gas and 13% of undiscovered oil thought to be located in the
Arctic (Gautier et al., 2009), increasing open water makes previously inaccessible areas of the Arctic available for oil and gas development and shipping (Harsem et al., 2015; Allison and Bassett, 2015). These developing-oil and gas extraction activities will-add pollutants, including particulate matter (PM), volatile organic compounds (VOCs), SO2, and NOx, to the Arctic atmosphere (Peters et al., 2011), thereby influencing Aretie climate. The Arctic aerosol population is characterized by a maximum mass loading
in the winter, due to transported pollutants from the mid-latitudes, and a minimum in the summer, when local sources, including sea spray aerosol, dominate (Quinn et al., 2002; Quinn et al., 2007). However, there is limited knowledge of aerosols produced within the Arctic, particularly in the context of changing emissions from both natural and anthropogenic sources (Arnold et al., 2016).

- Modeling by Peters et al. (2011) estimates that Arctic oil and gas extraction during 2004 15 contributed 47 kilotons (kt) of particulate matter (PM) emissions, with in the Arctic during 2004; 15 kt correspond to black carbon (BC) and, with 16 kt attributed to organic carbon (OC). The majority of eEmissions-primarily originated in western Russia (~41 kt in 2004 a year); activities within the Alaskan Arctic, primarily the Prudhoe Bay oil fields, contributed however, the United States (US) contributes an additional 6 kt during 2004 primarily from Prudhoe Bay (Peters et al., 2011), Prudhoe Bay is the third
- 20 largest oil field in the US and tenth largest gas field in the US by estimated production as of 2013 (EIA, 2015). The majority of the PM emitted by US Arctic oil and gas extraction sources (turbine gas combustion, diesel emissions from generators and vehicles, and flaring (Stohl et al., 2013)) in 2004 corresponded to contributed to the Arctic in 2004 from Prudhoe Bay was-BC (1.9 kt) and OC (2.0 kt) (Peters et al., 2011)., attributed to gas combustion within turbines, diesel emissions from generators and
- 25 vehicles, and flaring (Stohl et al., 2013). With new drilling operations opening due to reduced sea ice coverage, Peters et al. (2011) estimate US contributions increasing up to 10 kt of primary PM (including 3.3 kt BC and 3.5 kt OC) by 2030 and 17 kt of PM (including 5.3 kt BC and 5.7 kt OC) by 2050. In addition to directly emitted PM, drilling operations can emit aerosol precursors such as (NOx, SO2, and

VOCs), and alter oxidant levels, which can lead to the formation of secondary aerosol, as well as contribute to new particle formation (Peters et al., 2011; Volkamer et al., 2006; Roiger et al., 2015; Kolesar et al., 2017; Jaffe et al., 1995)., leading (Peters et al., 2011; Volkamer et al., 2006; Roiger et al., 2015; Kolesar et al., 2017).

- 5 BC is estimated to have a has a strong-warming effect on the Arctic atmosphere (e.g. Flanner, 2013; Sand et al., 2013a; Sharma et al., 2013; Bond et al., 2013; Flanner et al., 2007). -due to amplification from cloud and sea ice feedbacks (Flanner, 2013), with BC shown to contribute to melting of sea ice by decreasing snow and sea ice albedo (Ramanathan and Carmichael, 2008). 
[revised manuscript text omitted]

- 15 method of Dal Maso et al. (2002). The average condensation sink was 6 x 10-4-s-4, over an order of magnitude lower than typically observed at mid-latitude and boreal forest sites (e.g. Jung et al., 2013; Dal Maso et al., 2002; Kulmala et al., 2001). Based on the simulations by Fierce et al. (2015), accumulation of secondary species during transport for particles ~30-50 nm would take ~1-7 days if it was coagulation-dominated, which is likely due to the low amount of condensable species in the clean Arctic environment
- 20 (Ziemba et al., 2010). Therefore, particles of this size could be transported from Prudhoe Bay (average 21±7 hour transit time to Utqiagvik based on HYSPLIT backward air mass trajectories) without growing to larger diameters. Notably, Kolesar et al. (2017) previously observed Prudhoe Bay air masses preferentially exhibit particle growth, compared to Arctic Ocean air masses; therefore, it is clear the required precursors were not available for particle growth during the 2015 study.

**25 3.2 Single Particle Chemical Characterization**

Analysis of the individual particle  $(0.1 - 4.0 \,\mu\text{m})$  ATOFMS and CCSEM-EDX spectra resulted in the identification of five major single-particle types: sea spray aerosol (SSA), soot, organic carbon (OC),

biomass burning, and mineral dust (Figure 4). Detailed descriptions of particle-type mass spectra and classifications can be found in the supplemental information. SSA internally mixed with nitrate (NO2- [m/z -46] or NO3- [m/z -62] using with ATOFMS, N with EDX) and/or sulfate (SO3- [m/z -80] with ATOFMS, S with EDX) were sub-classified as partially aged SSA (Qin et al., 2012; Gard et al., 1998)

[revised manuscript text omitted]

2007; Wang et al., 2010).
Similar number fractions of fine mode soot particles were observed by CCSEM-EDX during both
Prudhoe Bay and Arctic Ocean periods (5 – 10% and 5 – 20%, by number, across 0.13 – 1 μm, respectively) (Figure 6). Though not statistically significant, ATOFMS-Soot was also identified increased soot by numberATOFMS during Prudhoe Bay periods (18 ± 14%) compared to Arctic Ocean periods (5

 $\pm$  2%). Identified by Cn+ clusters (C+ [m/z 12 [C+]7], C2+ [m/z 24 [C2+]7], C3+ [m/z 36 [C3+]7], etc) in ATOFMS spectra, soot). Elevated black carbon mass concentrations (up to 0.27 µg/m3) were also measured by the aethalometer during the Prudhoe Bay air mass observed on August 25 (Figure S5). Soot particles are primarily emitted through diesel combustion from heavy duty vehicles (Spencer et al., 2006) and ships (Ault et al., 2009). TheHowever, the majority of soot particles are is expected to be less than 100 nm in diameter and therefore not chemically characterized in this study. During the 2012 ACCESS campaign off the coast of Norway, Roiger et al. (2015) observed increased soot mass concentrations <80nm80 nm in diameter while sampling near oil and gas extraction facilities, consistent with the observed elevated ultrafine particlesparticle number concentrations in the present study when under

10 Prudhoe Bay air mass influence (Figure 2 and 3).

[revised manuscript text omitted]

---

## Author Response (AR2)

**acp-2017-453: Response to Editor Comments**

**We thank the editor very much for her helpful comments and suggestions. We provide below the original comments, shown in gray, and our responses, with specific revisions noted, in bold font.**

Page 5 line 5 "...background Arctic aerosol combustion." Do the authors mean aerosol composition?

> **This has been corrected to composition.**

Page 10 line 26: Was Kolesar et al (2017) based on one year or multiple years? Does the data analyzed in that work overlap with the data presented here? Regardless, please state the time period that the Kolesar et al. (2017) work covers.

> **Kolesar et al. (2017) was based on semi-continuous measurements between January 2008 and July 2015, and therefore that work unfortunately does not overlap with the data presented here. This sentence was revised as follows: "During semi-continuous measurements from January 2008 – July 2015, Kolesar et al. (2017), previously observed Prudhoe Bay air masses to preferentially exhibit particle growth, compared to Arctic Ocean air masses."**

Sect. 3.1 The 2015 measurements attributed to Prudhoe Bay (Fig. 2) are quite different from the 2008, 2009, 2013, an 2014 (Fig. 3) measurements. In particular, the average distribution between 20-30 nm is much larger in 2015 and lies beyond the 75th percentile in the multi-year measurements. Is there any indication as to what drives this difference?

> **One Prudhoe Bay air mass was observed during the August 2015 SMPS measurements, and unfortunately, without additional supporting information, it is not clear these ultrafine particle concentrations are higher than the previous (2008, 2009, 2013, 2014) Aug.-Sept. average. This discussion has been clarified on P10 L7-8, indicating the extent of the Prudhoe Bay measurements. However, even with these 2015 measurements being greater than the 75$^{th}$ percentile of the previous years, the observed trend holds true for higher ultrafine particle concentrations in the Prudhoe Bay air mass, compared to Arctic Ocean air masses.**

Page 12 line 6-8: Sierau et al. (2014) suggest options for a "missing" ATOFMS particle type other than pure ammonium sulfate such as organic particles of biological origin. Please expand the description of Sierau et al. (2014) to represent the discussion presented in that paper. Please address how that discussion relates to the results presented here. It may be worth considering moving this to Sect. 3.2.1.

> **While Sierau et al. (2014) does suggest organic particles of biological origin, CCSEM-EDX during the present study identified a sulfur particle type (not identified by ATOFMS) which is consistent with the presence of ammonium**

**sulfate or a similar sulfur-rich particle type. This is also consistent with previous ATOFMS studies which identified a missing particle type as ammonium sulfate (Wenzel et al., 2003; Spencer et al., 2008). The energy associated with a 266 nm two-photon ionization process (9.3 eV) is less than the lattice energy of ammonium sulfate (18.3 eV; Thomson et al., 1997), but greater than the ionization energy needed for atmospherically-relevant organics (less than 9 eV; Mysak et al., 2005), including expected organics of biogenic origin such as methanesulfonic acid (8.5 eV) and dimethyl sulfide (8.7 eV) (Lias et al., 2017). In addition, organic biological particles have been previously detected by ATOFMS (Gaston et al., 2011; Pratt et al., 2009). We also have no supporting evidence from the CCSEM-EDX data that the ATOFMS missed an organic biological particle type. However, we clarified P12 L15 to mention the potential missing organic particles suggested by Sierau et al. (2014), and we expanded P12 L16-17 linking the identification of sulfur particles by CCSEM-EDX to the "missing" ATOFMS particle type, in addition to the previous added discussion of the impact on the number fractions of ATOFMS particle types (P12 L16-21).**

Page 16 line 4-5: For the CCSEM-EDX data the sulfur-rich particles are also important and greater than fresh SSA in some of the size bins from 0.13-0.4 um.

**We have clarified this sentence to read that "...fresh SSA was a major contributor..." instead of "...fresh SSA was the major contributor...".**

Page 16 line 17: "...potential to grow..." It may be prudent to re-emphasize how growth was not observed in this work.

**This sentence now reads, "Though not observed in the present study, these transported particles have the potential to grow (Kolesar et al., 2017) and serve as CCN...".**

Figure 2: Please include 10 nm on the x-axis (as is done in Figure 3)

**Figure 2 only has data down to 14 nm, compared to data down to 10 nm in Figure 3. This has been clarified in both figure captions, as well as the methods for each individual instrument.**

Figure 6: Please consider including a histogram showing the number of particles analyzed in each bin. Alternatively, this may be provided in the supplement.

**We have added this histogram to the supplemental information as Figure S6, and added the following statement to the caption of Figure 6: "A histogram for the number of particles analyzed in each bin can be found in the Supplemental Information (Figure S6)."**

Supplement

Page 1 line 16: -93/95 should be NaCl2-

**This has been corrected.**

[Figure]

 **Figure S1.** Wind rose from August 21–September 30, 2015 measured at the NOAA Barrow Observatory. Wind speed is binned by 2 m/s, and wind direction is binned by 20 degrees, with the radial axes representing the fraction of the study under those wind conditions.

120

[Figure]

**Figure S2.** Aerosol size-resolved number concentrations (mobility diameter) measured by the SMPS from August 21-September 20, 2015. Identified air mass source regions, determined based on wind direction and backward air mass trajectories, are labeled and divided by white lines in the time series. Periods lacking data are indicated in gray. Total particle (0.013 – 746 nm) number concentrations are also shown.

125

[Figure]

130 **Figure S3.** Median, as well as 25th and 75th percentile, particle size distributions, measured by SMPS, during Prudhoe Bay and Arctic Ocean influenced air masses from August 21–September 20, 2015.

[Figure]

**Figure S4.** S/Na, N/Na, Cl/Na mole ratios of individual SSA (top) and fraction of OC particles (bottom) containing S, N, and/or Cl, measured by CCSEM-EDX for Arctic Ocean and Prudhoe Bay influenced air masses. Size bins with less than 25 particles are not displayed.

[Figure]

**Figure S5.** Aerosol size-resolved number concentrations (mobility diameter) measured by the SMPS
during Prudhoe Bay air mass influence on August 24-25, 2015. Black carbon mass concentrations
measured by the aethalometer are overlaid in white.

[Figure]

Figure S6. Numbers of particles, per particle projected diameter size bin, analyzed by CCSEM-EDX for
(A) Arctic Ocean and (B) Prudhoe Bay air mass samples.